# Policy Design in Long-run Welfare Dynamics

Jiduan Wu[1,2], Rediet Abebe[1,3,*], Moritz Hardt[1,*], and Ana-Andreea Stoica[1,*]

[1]*Max Planck Institute for Intelligent Systems, Tübingen and Tübingen AI Center*
[2]*Department of Computer Science, ETH Zürich*
[3]*ELLIS Institute, Tübingen*

## Abstract

Improving social welfare is a complex challenge requiring policymakers to optimize objectives across multiple time horizons. Evaluating the impact of such policies presents a fundamental challenge, as those that appear suboptimal in the short run may yield significant long-term benefits. We tackle this challenge by analyzing the long-term dynamics of two prominent policy frameworks: Rawlsian policies, which prioritize those with the greatest need, and utilitarian policies, which maximize immediate welfare gains. Conventional wisdom suggests these policies are at odds, as Rawlsian policies are assumed to come at the cost of reducing the average social welfare, which their utilitarian counterparts directly optimize. We challenge this assumption by analyzing these policies in a sequential decision-making framework where individuals' welfare levels stochastically decay over time, and policymakers can intervene to prevent this decay. Under reasonable assumptions, we prove that interventions following Rawlsian policies can outperform utilitarian policies in the long run, even when the latter dominate in the short run. We characterize the exact conditions under which Rawlsian policies can outperform utilitarian policies. We further illustrate our theoretical findings using simulations, which highlight the risks of evaluating policies based solely on their short-term effects. Our results underscore the necessity of considering long-term horizons in designing and evaluating welfare policies; the true efficacy of even well-established policies may only emerge over time.

## 1 Introduction

An important application of sequential decision making is the problem of promoting long-run social welfare through a sequence of targeted interventions in a population. Policies for this problem face a two-fold challenge. On the one hand, they must be effective at optimizing the long-term objective. On the other hand, they must appeal to the political and normative expectations of policy makers. In particular, simple policies supported by established moral and political arguments are desirable. Two families of policies have been particularly influential in the context of Western welfare programs. One targets individuals of largest immediate welfare gain. The other targets those most seriously in need. While the former derives from utilitarian moral principles, the latter is associated with Rawl's theory of justice. Many scholars, however, have criticized Rawlsian policy for its presumed failure to maximize social welfare.

Indeed, there is no obvious reason why allocating resources to those of lowest welfare should also maximize average welfare in the long run. In this work, we study a stochastic dynamic model of long-term welfare in a population. Surprisingly, under reasonable assumptions on the welfare dynamics, Rawlsian policy turns out to outperform an idealized utilitarian policy that chooses the individual of largest treatment effect at each step. This is the case even though the Rawlsian policy is suboptimal on a short-term horizon.

Although our motivation is social welfare, our results hold a broader lesson for sequential decision making. Simple policies can be highly effective, but their long-run efficiency may not be apparent on a short time horizon.

---

*Alphabetical order.

## 1.1 OUR CONTRIBUTIONS

We propose a multi-agent stochastic dynamical model to describe long-run welfare in a population of individuals. Our model draws from classical economic theory of industrial project management, extending so-called *attention allocation policies* (Radner & Rothschild, 1975) into social policies.

In our model, each individual $i$ has a welfare level $U_i(t)$ at each timestep $t$. At each timestep, a social planner allocates an intervention to one or more of $N$ agents using some policy $\pi$. The welfare values evolve according to a stochastic dynamical system. Absent an intervention, an individual's welfare decays in expectation according to a function $g_i(U_i(t)) > 0$. When the social planner allocates an intervention to an individual, however, the individual's welfare increases in expectation according to a function $f_i(U_i(t)) > 0$. We are interested in comparing Rawlsian and utilitarian policies based on the *long-term social welfare* they achieve, i.e. the asymptotic individual welfare increase, defined as $\lim_{t\to\infty} (U_i(t) - U_i(0))/t$, averaged over all individuals.

We make two substantive assumptions about welfare dynamics. The well-known Matthew effect (Merton, 1968; Rigney, 2010), or "rich-get-richer" while "poor-get-poorer" dynamic, suggests that inequality amplifies over time. We capture this effect by assuming that the return function $f_i(\cdot)$ is increasing with welfare, while the decay function $g_i(\cdot)$ decreases with welfare. The other assumption is a uniform boundedness assumption: the bounds of the return on intervention and decay functions are the same for all individuals. In other words, no individual can achieve a highest/lowest possible level of return or decay that is much higher or much lower than anyone else.

Under these assumptions, we find a sufficient condition for comparing policies. This condition states that a policy can, in principle, avoid the decay of any individual's welfare below 0. We call this a *survival condition* and note that it rests on the functional form and bounds of the return and decay functions. Informally, our main result shows:

> Under the survival condition, Matthew effect, and uniform boundedness, a Rawlsian policy will achieve better long-term social welfare than a utilitarian policy almost surely.

We complement this result by characterizing a condition in which the reverse is true: under a so-called "ruin condition" (when a policy cannot prevent an individual's unbounded welfare decay), a utilitarian policy will achieve better long-term social welfare than a Rawlsian policy almost surely.

To prove our results, we present a series of theoretical results that characterize in closed form the rate of growth of individual welfare under Rawlsian and utilitarian policies (Sections 3 and 4). Our proof extends the elegant argument of Radner & Rothschild (1975), who studied a fully homogeneous case in which the return and decay functions are constant terms. This generalization in turn requires a non-trivial departure from the original proof including a variant of Lundberg's classical inequality for submartingale processes. The proof may be of independent interest for similar problems arising in sequential decision making and reinforcement learning.

We illustrate our theoretical results by simulating our model with initial conditions drawn from real data from the Survey of Income and Program Participation (SIPP) of the U.S. Census Bureau in Section 5. We see a delayed effect of a Rawlsian policy, noting that it obtains lower social welfare in the short-term, yet quickly converges to a higher social welfare value than the utilitarian policy. We highlight limitations of our work and directions for future study in Section 6. Finally, we discuss potential extensions of our work (e.g., when the functions $f_i, g_i$ violate the uniform boundedness assumption) in the appendix.

## 1.2 RELATED WORK

Welfare-based social policies have a long history in economics research (Sen, 1979; Kaplow & Shavell, 2000; Adler, 2011). Although Rawlsian principles are based on distributive justice and egalitarian goals (Harsanyi, 1975; Blau & Abramovitz, 2010), debates remain regarding their efficiency as compared to utilitarian policies (Arrow, 1973; Sen, 1976). The direct comparison between Rawlsian and utilitarian policies generally remains an open area of research, with some empirical and model-based comparisons made in the context of optimal taxation policies (Atkinson, 1995) and income inequality (Mongin & Pivato, 2021).

We build on the model proposed by Radner & Rothschild (1975) in the context of industrial project management, which analyzes the behavior of the system under different attention allocation mechanisms. We generalize and re-purpose their model by equipping it with various functional forms of the return and decay functions that capture societal behaviors and analyzing several additional policies. Our modeling choices include the Matthew effect (Merton, 1968): individuals with higher level of welfare may benefit the most from interventions ("rich-get-richer"), whereas individuals with low wealth experience more severe income shocks absent any interventions from the social planner ("poor-get-poorer"). Such effects have been documented in the context of economic inequality (Rigney, 2010; Stiglitz, 2012) and optimal taxation policy for reducing societal inequality (Atkinson, 2015).

Closely related to our work are recent modeling frameworks for wealth fluctuations and policy design. Two recent papers develop algorithms for selecting the optimal candidates for intervening, subject to different objectives: Abebe et al. (2020) analyze two policy objectives in a population that undergoes income shocks and proposes algorithms for allocating subsidies optimally; their objectives aim to minimize the probability of ruin for any given individual. Arunachaleswaran et al. (2022) analyze the theoretical complexity and give approximation algorithms for the optimal selection of candidates under a social welfare and a Rawlsian objective, considering a transition matrix of welfare states. In addition, Heidari & Kleinberg (2021) study the optimal policy for allocating interventions in a population with two welfare states (advantaged and disadvantaged), over a finite time horizon. Acharya et al. (2023) study the effect interventions in a welfare-based dynamic system with feedback loops in societal inequality. Their interventions include allocating subsidies to those among most in need, without a comparison between different types of policies on the social welfare. In contrast, we study the effect of different policies in the long-run, formulating a sufficient condition for a Rawlsian policy to achieve better welfare than a utilitarian policy.

A related line of work focuses on reinforcement learning algorithms for deriving optimal policies. In particular, Zheng et al. (2020) propose a framework for a integrating AI into two-level optimization problem in the context of optimal taxation policy, with subsequent work improving the generality of the model (Curry et al., 2023). Offline and online algorithms have been proposed for finding optimal policies with fairness considerations (Zimmer et al., 2021; Zhou, 2024) as well as in contexts with strategic agents Liu et al. (2022). The problem of optimal policy selection can be tackled using a continuous-state MDP under the average-reward criteria, with early works considering bounded reward rates (Doshi, 1976) and subsequent extensions that do not require boundedness (Guo & Rieder, 2006). These works find theoretical guarantees for the *existence* of optimal policies, convergence rates, as well as optimality gaps. Often, such works do not find tractable, closed-form solutions for the optimal policy, but rather build heuristics with theoretical guarantees that can closely approximate an optimal policy.

Finally, the problem of allocating resources through objectives such as a maximin rule includes lines of work in fair division (Procaccia & Wang, 2014) as well as machine learning, often as a constraint in a larger optimization problem (Binns, 2018; Heidari et al., 2019). Other works have studied Rawlsian principles under a finite time horizon (Dwork et al., 2012; Zafar et al., 2017; Diana et al., 2021) or as a static optimization problem (Chen & Hooker, 2020; Stark et al., 2014). Some works have studied the long-term effect of fair algorithms in the context of hiring (Hu & Chen, 2018) and resource-allocation (Liu et al., 2018). Kube et al. (2019) and Azizi et al. (2018) offer data-driven approaches for optimal assigning subsidies to individuals who experience homelessness; their approach uses a prioritization scheme that aims to minimize the probability of an individual to re-enter homelessness, based on an automated prediction.

## 2 A MODEL OF WELFARE DYNAMICS AND SOCIAL POLICIES

**Preliminaries.** Consider $N$ individuals indexed by $i = 1, \ldots, N$. Each individual $i$ has a welfare value of $U_i(t)$ at each timestep $t \geq 0$. The initial welfare values $U_i(0)$ are drawn from a distribution (e.g. a capped normal distribution; different choices of the initial distribution do not change our results). Here, welfare may represent the household income level, expenditure, monthly income, or other variables that define individual welfare.

An intervention at time $t$ is defined through a vector $\boldsymbol{a}(t) \coloneqq (a_i(t))_i$, where an amount of $a_i(t)$ budget is allocated to individual $i$ by the social planner. The exact decision of *who* receives an

amount of budget and *how much* they receive is decided by the social planner through a *social policy*. The social planner has a budget $M$ for allocating interventions at every timestep $t \geq 0$: $\sum_i a_i(t) = M$, for $M \in \mathbb{N}, 1 \leq M \leq N$, and $0 \leq a_i(t) \leq 1$. In this first analysis, we consider the case when the social planner can only allocate an integer unit to each individual, so $a_i(t) \in \{0, 1\}$.

## 2.1 A DYNAMIC MODEL OF WELFARE FLUCTUATIONS.

Absent any intervention, we assume that the welfare of individuals fluctuates at every timestep according to a function $g_i : \mathbb{R} \to \mathbb{R}_+$, defined as a function of the welfare value for each individual $i$. We denote the function $g_i(\cdot)$ as the **decay function**, capturing the welfare decrease in natural conditions (e.g., income shocks due to accidents, economic conditions, natural disasters).

In contrast, we model the impact of interventions on individuals' welfare at each timestep through a function $f_i : \mathbb{R} \to \mathbb{R}_+$, defined for all individuals $i$. We refer to $f_i(\cdot)$ as the intervention **return function**, capturing the effect of intervening on an individual (e.g., a new job through an employment program, social benefits, cash transfers). Let $\mathcal{F}_t$ be a $\sigma-$algebra denoting the space of events up to time step $t$. We model the rate of change of individual welfare between different timesteps under interventions as:

$$\mathbb{E}[U_i(t+1) - U_i(t) \mid \mathcal{F}_t] = a_i(t) \cdot f_i(U_i(t)) - (1 - a_i(t)) \cdot g_i(U_i(t)) \qquad (1)$$

Treatment ($a_i(t) = 1$) in our model has two effects. On the one hand, the treated individual realizes the return $f_i(U_i(t))$. On the other hand, the treated individual avoids the decay $-g_i(U_i(t))$. The *individual treatment effect* of allocating an intervention to individual $i$ at time $t$ therefore corresponds to the expression

$$f_i(U_i(t)) + g_i(U_i(t)) \,.$$

Note that this quantity varies both in time and by individual. Conceptually, targeted individuals have a positive return, whereas non-targeted individuals suffer a decay in their welfare.

## 2.2 SOCIAL POLICIES

A *policy* $\pi$ selects an individual for treatment at each step. This corresponds to setting the coefficients $\{a_i(t)\}$ at each timestep $t$. We restrict our attention to policies that allocate $M$ units of resources to $M$ individuals with each individual receiving exactly one unit at each time step. Let $\text{Top}_s(S)$ denote the set of $s$ largest elements of set $S$.

A natural utilitarian policy is the one that chooses the individual of largest treatment effect. We call this the **max-fg policy**:

$$a_i(t) = \begin{cases} 1, & i \in \text{Top}_M \left( \{f_k(U_k(t)) + g_k(U_k(t))\}_{k=1}^N \right) , \\ 0, & \text{otherwise.} \end{cases} \qquad \text{(max-fg)}$$

Note that this policy requires full information about individual treatment effects at each time step. This may be an unrealistic requirement in many applications. We call this the **max-U policy**:

$$a_i(t) = \begin{cases} 1, & i \in \text{Top}_M \left( \{U_k(t)\}_{k=1}^N \right) , \\ 0, & \text{otherwise.} \end{cases} \qquad \text{(max-U)}$$

The max-U policy is *welfare-based* and requires only welfare measurements for its implementation. This utilitarian welfare-based policy directly contrasts with the Rawlsian policy that chooses the individual of minimum welfare at each step. We call this the **min-U policy**:

$$a_i(t) = \begin{cases} 1, & i \in \text{Top}_M \left( \{U_k(t)\}_{k=1}^N \right) , \\ 0, & \text{otherwise.} \end{cases} \qquad \text{(min-U)}$$

Radner & Rothschild (1975) studied these policies under the names "putting out fires" for min-U and "staying with a winner" for max-U with $M = 1$.

We explore a variation of the utilitarian policy that only uses knowledge of the intervention return functions $f_i(\cdot)$, i.e. the policy will allocate a unit of effort to the individual with the highest intervention return:

$$a_i(t) = \begin{cases} 1, & i \in \text{Top}_M\left(\{f_k(U_k(t))\}_{k=1}^N\right), \\ 0, & \text{otherwise.} \end{cases} \qquad \text{(max-f)}$$

We call this **max-f**. In contrast to max-fg, the max-f policy requires only partial information about the interventions, only measured through the return on interventions which may be less costly to measure. By analogy, we consider a variant of the Rawlsian policy here that only use knowledge of the decay functions $g_i(\cdot)$. That is, the **max-g** policy will allocate a unit of effort to the individual with the highest decay:

$$a_i(t) = \begin{cases} 1, & i \in \text{Top}_M\left(\{g_k(U_k(t))\}_{k=1}^N\right), \\ 0, & \text{otherwise.} \end{cases} \qquad \text{(max-g)}$$

*Tie-breaking rule:* Among individuals with the same welfare, we favor the one with the lowest index $i \in [N]$. This applies to all policies. For the policies that use the treatment effect information, max-f and max-fg, we break the tie in favor of the individual with the lowest index. For the max-g policy, among individuals with the same $g_i$ value, we break the tie in favor of the individual with the lowest welfare value, arguing that this best captures a Rawlsian principle. When max-g prioritizes the lowest index individual, results do not qualitatively change (see Appendix E, Figure 4).

**Policy goal.** The goal of a policy is to promote long-term social welfare. Our main results focus on the long-term social welfare comparison of Rawlsian and utilitarian policies. We capture long-term social welfare as the average asymptotic welfare gain among individuals, defined as follows.

**Definition 1** (Long-term social welfare)**.** *The long-term average social welfare induced by policy $\pi$ on a population of $N$ individuals is defined as*

$$\bar{R}_\pi := \frac{1}{N}\sum_{i=1}^N R_i, \quad R_i := \lim_{t\to\infty}\frac{U_i(t) - U_i(0)}{t}. \qquad (2)$$

*where $R_i$ defines the rate of growth of individual $i$, asymptotically.*

Note the welfare level $U_i(t)$ depends on the policy $\pi$, as $\pi$ determines $\boldsymbol{a}(t)$ at every timestep, and therefore the subsequent $U_i(t+1)$ through the model described in Equation 1.

### 2.3 MODELING CHOICES

The comparison between Rawlsian and utilitarian policies depends on an important condition, called a 'survival' condition. Survival means that no individual in a population will obtain negative welfare. The survival condition is necessary and sufficient to obtain a positive probability of survival for all individuals under some policy, as noted by Radner & Rothschild (1975). Such a policy only exists under the survival condition, and in fact, Rawlsian policies are examples as we will show later in Section 3. This is a sufficient condition for comparing policies in the long run. Formally, the survival condition can be stated in terms of a weighted sum of the $f_i(\cdot)$ and $g_i(\cdot)$ function bounds (assuming those exist):

**Assumption 1** (Survival condition)**.** *We assume $\bar{\zeta}((f_1^-,\ldots,f_N^-),(g_1^+,\ldots,g_N^+)) > 0$ where $\bar{\zeta} : \mathbb{R}^{2N} \to \mathbb{R}$ is defined as*

$$\bar{\zeta}((x_1,\ldots,x_N),(y_1,\ldots,y_N)) := \left(M - \sum_{i=1}^N \frac{y_i}{x_i + y_i}\right)\left(\sum_{j=1}^N \frac{1}{x_j + y_j}\right)^{-1}, \qquad (3)$$

*and $f_i^+ := \sup f_i(x) > 0$, $f_i^- := \inf f_i(x) > 0$, $g_i^+ := \sup g_i(x) > 0$, $g_i^- := \inf g_i(x) > 0$.*

Next, we formally state the modeling conditions that capture a Matthew effect, as motivated in the introduction, as well as a uniform boundedness condition.

**Assumption 2** (Modeling conditions). *(a). (Rich-get-richer) For $i = 1, \ldots, N$, we assume that the function $f_i(x)$ is non-decreasing.*

*(b). (Poor-get-poorer) For $i = 1, \ldots, N$, we assume that the function $g_i(x)$ is non-increasing.*

*(c). (Uniform boundedness) For $i = 1, \ldots, N$, we assume $f_i^- \equiv f^-$, $f_i^+ \equiv f^+$, $g_i^- \equiv g^-$, $g_i^+ \equiv g^+$ for constants $f^-$, $f^+$, $g^-$, $g^+$.*

We note that this assumption does not require that the functions $f_i, g_i$ be the *exact same* for all individuals, but rather just their limits.

Finally, in addition to the two assumptions described above, our results require some standard regularity conditions, formalized below. Denote the welfare variation between two consecutive timesteps by $Z_i(t+1) := U_i(t+1) - U_i(t), \forall i \in [N]$. We note that $U_i(t)$ and $Z_i(t)$ are random variables with respect to a stochastic process of welfare fluctuations over time (e.g., income shocks).

**Assumption 3** (Regularity conditions). *Consider a probability space $(\Omega, \mathcal{F}, \mathbb{P})$, where $\Omega$ is the space of possible outcomes of welfare levels, $\mathcal{F}$ is a $\sigma-$algebra denoting the space of events, and $\mathbb{P} : \mathcal{F} \to [0,1]$ is a probability measure function. We assume the following properties:*

*(a). The welfare random variable $U_i(t)$ is $\mathcal{F}_t$-measurable for $\forall i \in [I]$, $t \in \mathbb{N}^*$, for $\mathcal{F}_0 \subset \mathcal{F}_1 \subset \cdots$ an increasing sub $\sigma$-field of $\mathcal{F}$.*

*(b). The variation random variable $Z_i(t+1)$ is integer-valued, mutually independent (given $\boldsymbol{a}(t)$), and uniformly bounded, i.e. $|Z_i(t+1)| \leq b$, $\forall i \in [I], t \in \mathbb{N}$ for some constant $b > 0$.*

*(c). There exist constants $z^*, l > 0$ with $0 < l < 1$ s.t. $\mathbb{P}(Z_i(t+1) \geq z^* \mid \mathcal{F}_t) \geq l$, $\mathbb{P}(Z_i(t+1) \leq -z^* \mid \mathcal{F}_t) \geq l$ for any $i \in [N]$, any $\mathcal{F}_t$, $\forall t \geq 0$.*

## 3 POLICY COMPARISONS IN TERMS OF LONG-TERM SOCIAL WELFARE

Our main result compares the long-term social welfare of Rawlsian and utilitarian policies, under the natural behavioral model of welfare fluctuations described in Section 2.1.

**Theorem 1** (Main result). *For a population of $N$ individuals whose welfare $(U_i(t))_i$ fluctuates according to the model in equation 1, under regularity, modeling, and survival conditions (Assumptions 1, 2, 3), a Rawlsian policy will achieve better long-term social welfare than a utilitarian policy:*

$$\bar{R}_{\text{Rawlsian}} \geq \bar{R}_{\text{utilitarian}} \quad a.s.$$

*where the Rawlsian and utilitarian policies are defined in the same informational contexts, i.e.* $(\min\text{-}U, \max\text{-}U), (\max\text{-}g, \max\text{-}f), (\max\text{-}g, \max\text{-}fg)$.

*Proof sketch.* The proof of Theorem 1 includes a series of results on the individual rates of growth for different policies. First, we compute the individual rate of growth under the Rawlsian policy to be equal for all individuals (Theorem 3). The survival condition implies the existence of a policy that prevents any individual's welfare from decaying below $0$. In fact, it implies something even stronger: under survival, a Rawlsian policy can 'lift' everyone's welfare unboundedly: $\lim_{t \to \infty} \min_i U_i(t) = \infty$ almost surely. This helps us show that the welfare gap between any two individuals vanishes asymptotically, obtaining the same individual rates of growth for all individuals. In contrast, a utilitarian policy tends to fixate on a single individual and repeatedly allocate an intervention to him, while ignoring the rest of the population. We show this formally in Theorem 4: we leverage a generalization of Lundberg's inequality for submartingale processes to lowerbound the probability that a utilitarian policy repeatedly allocates interventions to the *same* high-welfare individuals. Finally, the individual rates of growth and uniform boundedness allow us to compute and compare the long-term social welfare under different policies, see Corollaries 1, 2. Essentially, a Rawlsian policy obtains better social welfare in the long-run than utilitarian policies as long as $\lim_{x \to +\infty} g_i(x) \leq \lim_{x \to -\infty} g_i(x)$, which is true by our "poor-get-poorer" modeling condition. It is noteworthy that the result holds regardless of the variation of our policies: whether the social planner has knowledge of $(f_i)_i, (g_i)_i$ or not, the policy comparison remains the same under our modeling conditions. Detailed proofs for all results can be found in Appendix A. $\square$

In cases where the survival condition does not hold, we find a natural complement for our theory: we define a "ruin condition" as a state of the model in which no policy can prevent all individuals from decaying below 0. Our theory under survival naturally extends for this ruin condition, showing that a utilitarian policy will achieve better long-term social welfare (see Appendix B for the formal theory).

**Theorem 2** (Policy comparison under a ruin condition). *For a population of $N$ individuals whose welfare $(U_i(t))_i$ fluctuates according to the model in equation 1, under regularity, modeling, and ruin conditions (Assumptions 2, 3, 4), a utilitarian policy will achieve better long-term social welfare than a Rawlsian policy:*

$$\bar{R}_{\text{Rawlsian}} \leq \bar{R}_{\text{utilitarian}} \quad a.s.$$

*where the Rawlsian and utilitarian policies are defined in the same informational contexts, i.e.* $(\text{min-U}, \text{max-U}), (\text{max-g}, \text{max-f}), (\text{max-g}, \text{max-fg}).$

## 4 INDIVIDUAL WELFARE RATE OF GROWTH UNDER DIFFERENT POLICIES

In this section, we characterize in closed form the rate of growth of welfare under different policies for all individuals, that is, proving that $R_i = \lim_{t \to \infty}(U_i(t) - U_i(0))/t$ converges to closed-form solutions for all $i \in [N]$. We then compute the long-term average social welfare achieved by all policies and compare them against a baseline defined by a random allocation policy.

### 4.1 INDIVIDUAL WELFARE UNDER THE RAWLSIAN POLICY

**Theorem 3.** *Under regularity (Assumption 3), modeling conditions (Assumption 2.(a),(b)), and the survival condition (Assumption 1), a Rawlsian policy $\pi \in \{\text{min-U}, \text{max-g}\}$ leads to the following closed-form solution of the individual rates of growth:*

$$R_i = \bar{\zeta}((f_1^+, \dots, f_N^+), (g_1^-, \dots, g_N^-)), \quad i = 1, \dots, N, \quad a.s.$$

**Corollary 1.** *With the addition of the uniform boundedness condition from Assumption 2.(c), we can simplify the individual rates of growth, obtaining the long-term social welfare value for the Rawlsian policy:*

$$\bar{R}_{\text{min-U}} = \bar{R}_{\text{max-g}} = \frac{M}{N}f^+ - \frac{N-M}{N}g^- \quad a.s.$$

*Proof sketch.* Under the survival condition, we prove that the minimum welfare level will be lifted unboundedly over time. We model the welfare gap between the treated and untreated individuals and show that this gap vanishes almost surely by applying the law of large numbers. We conclude by adapting a convergence argument first introduced by Radner & Rothschild (1975), obtaining that a Rawlsian policy achieves the same long-run welfare of everyone under our modeling conditions. □

### 4.2 INDIVIDUAL WELFARE UNDER THE UTILITARIAN POLICIES

**Theorem 4.** *Under regularity (Assumption 3) and modeling conditions (Assumption 2.(a),(b)) and as long as $f_i(\cdot) + g_i(\cdot)$ is increasing for all $i \in [N]$,[*] a utilitarian policy $\pi \in \{\text{max-U}, \text{max-fg}, \text{max-f}\}$ leads to the following closed-form solution of the individual rates of growth:*

$$R_i = \begin{cases} f_i^+, i \in J, \\ -g_i^+, i \notin J, \end{cases} \quad a.s.$$

*where $J$ is a set of $M$ random variables with values in $[N]$ whose exact value depends on $U(0), (f_i(\cdot))_i,$ and $(g_i(\cdot))_i$. In other words, exactly $M$ individuals achieve an asymptotic rate of growth equal to $f_i^+$, whereas all others achieve $-g_i^+$.*

---

[*]This assumption states that the return from an intervention should, in principle, be higher than the shock experienced by an individual absent intervention. It is only needed for the max-fg policy, since it is the only one using knowledge of both the return and decay functions.

**Corollary 2.** *With the addition of the uniform boundedness condition from Assumption 2.(c), we can simplify the individual rates of growth, obtaining the social welfare value*

$$\bar{R}_{\text{max-U}} = \bar{R}_{\text{max-f}} = \bar{R}_{\text{max-fg}} = \frac{M}{N}f^+ - \frac{N-M}{N}g^+ \quad a.s.$$

*Proof sketch.* For the individuals chosen by the utilitarian policy at a time $t_0$, we upperbound the probability of an individual obtaining negative welfare at a finite point in time (a variable that we model as a submartingale), for any $\mathcal{F}_t$ and individual $i$. We make use of a generalized Lundberg's inequality (Lundberg, 1903; Cramér, 1959; Moriconi, 1986) for submartingales, for which we provide an adapted version for our model and a new proof. We then use it to show that the probability that it will be chosen again afterwards ($a_i(t) \equiv 1, t \geq t_0$) is lower-bounded by some positive constant. Asymptotically, the probability of $M$ individuals being targeted by a utilitarian policy approaches 1, and hence we obtain the asymptotic convergence of the individual rates of growth under our modeling conditions. □

**Random policy.** Finally, we compare our policies with a baseline policy that randomly chooses $M$ individuals to allocate an intervention at every timestep.

**Theorem 5.** *Under regularity, modeling, and survival conditions (Assumptions 1, 2, 3), the random policy leads to the following closed-form solution of the individual rates of growth and long-term social welfare:*

$$\bar{R}_{\text{random}} = R_i = \frac{M}{N}f^+ - \frac{N-M}{N}g^-, \quad i = 1, \dots, N, \quad a.s.$$

*Proof sketch:* Under the assumption of uniform boundedness, we may lowerbound the rate of welfare increase at every timestep by a positive quantity, $\mathbb{E}[U_i(t+1) - U_i(t) \mid \mathcal{F}_t] \geq \frac{M}{N}f_i^- - (1 - \frac{M}{N})g_i^+ > 0$ under the random policy. This allows us to show that, in the limit, the welfare value of every individual will increase unboundedly. At the same time, since individuals are chosen randomly at each time, the welfare gap between individuals converges to 0, just like in the proof of Theorem 3. We follow a similar proof structure henceforth, detailed in Appendix A. □

**Policy comparison:** Our results show that Rawlsian and random policies will achieve better long-term social welfare than utilitarian policies under the aforementioned conditions. This concludes our argument for the main result in Theorem 1. Furthermore, our subsequent results show that the comparison holds no matter the informational context (whether the social planner uses only welfare information in defining policies, or also has access to the treatment effect through $f_i, g_i$). Furthermore, when the survival condition is not satisfied, we find a complementary condition under which a policy reversal occurs. We provide a formal theory for this result in Appendix B. We extend our results to include different functional forms for the treatment effect function (Appendix D) and allocate proportional interventions at each timestep (Appendix F). We explore different combinations of monotonicities of return/decay functions through simulations in Appendix G.

## 5 ILLUSTRATION OF THEORETICAL RESULTS AND MODELING CHOICES

We illustrate our theoretical results through simulations using a real-world dataset. We compare the average social welfare under a finite time horizon for all proposed policies.

### 5.1 SIMULATIONS OF POLICIES ON REAL DATA UNDER FINITE TIME HORIZON

We use data collected from the Survey of Income and Program Participation (SIPP) (Bureau, 2023), which is a longitudinal survey of households in the U.S. containing variables related to economic well-being such as income, employment, etc. Among numerous indices, we use the income variable as a proxy for the initial individual welfare level, $(U_i(0))_i$. We group the whole population $39,720$ into 13 bins and treat every 200 samples as one individual, and every $\$1,000$ as one welfare unit in our model. In the end, we obtain a population of 206 individuals. We simulate an instance of the general model from equation 1 with Gaussian noise, specified as:

$$U_i(t+1) - U_i(t) = a_i(t) \cdot f_i(U_i(t)) - (1 - a_i(t)) \cdot g_i(U_i(t)) + \xi_i(t), \quad \forall t \geq 0, i \in [N]. \quad (4)$$

where $\{\xi_i(t)\}_{i,t} \sim \mathcal{N}(0, \sigma^2)$ and capped within uniform bounds, for some noise parameter $\sigma$. We generate homogeneous bounds $f^-$, $f^+$, $g^-$, $g^+$, and then generate the functions $f_i(\cdot)$, $g_i(\cdot)$ as segment linear functions. Our results are averaged over 100 draws, reporting standard deviation in the error bands. See Appendix C for further experimental details.

We measure social welfare at timestep $t$ as the individual growth rate up to time $t$ averaged over all individuals (equation 2 up to time $t$).

The average social welfare (solid lines) converges to the theoretical expected welfare (dashed lines) for all policies (Figure 1). Furthermore, Rawlsian policies (min-U and max-g) have a lower short-term social welfare than utilitarian policies (max-U, max-f, max-fg). After a few hundreds timesteps, this trend is reversed, showing convergence to the theoretical social welfare value. Rawlsian policies achieve better welfare than utilitarian policies in the long-run, which is implied by Theorem 1. The random policy behaves similarly to the Rawlsian policy (as Theorem 5 would suggest), yet with a slower convergence rate. This disadvantage vanishes as the budget $M$ increases, as indicated by Figure 1b. Figure 2 in Appendix B illustrates the finite time horizon under the ruin condition, showcasing a reversal of the Theorem 1 result (for a formal statement, see Theorem 2). We also showcase the complexity of evaluating policies in heterogeneous cases (i.e., where the bounds of the return and decay functions $f_i, g_i$ are non-uniform) in Appendix H.

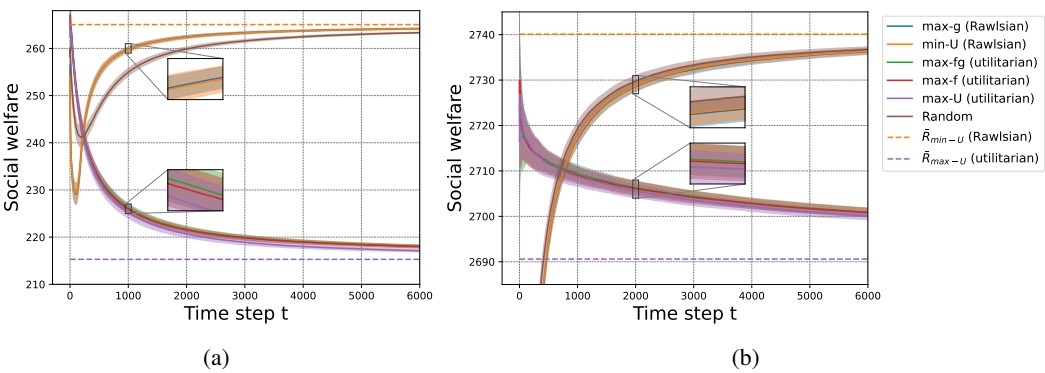

Figure 1: Social welfare as the finite-time growth rate averaged over all individuals, for all policies (solid lines), as well as theoretical expected growth rate, asymptotically (dashed lines) for budget $M = 1$ (a) and $M = 10$ (b).

All simulations are ran on commodity hardware, using Python 3.8. All code and data used in our simulations is available in this repository.

## 6 DISCUSSION

The problem of optimal policy design remains highly relevant, as several countries continue to implement changes in their social benefits allocation schemes. A prominent example is Austria, which has shifted from a welfare state approach that targeted those most in need to an "inactivity trap" approach that targets those most likely to (re)enter the labor market (Allhutter et al., 2020; Christl & De Poli, 2021; International). In 2019, Austria introduced the New Social Assistance policy that reduced benefits to individuals with low language skills or larger number of dependents. In 2020, it introduced algorithmic profiling by first predicting individuals' probability of re-entering the labor market, and second by offering most support to those with intermediate chances. Both such policies have the purpose of shifting support from the most in need to those with the highest chance of benefiting from such support, measured through their integration in the labor market in the near future. In essence, such policies have shifted from a Rawlsian approach to social welfare (Bufacchi & Garmise, 1995) to a more utilitarian view of social benefits (Christl & De Poli, 2021).

Our work demonstrates that choosing the right policy framework is subtle. In particular, our results motivate the necessity of long-run welfare comparisons of policies that a short-term analysis will necessarily miss. Whereas on the short-term horizon a utilitarian policy prevails, it can result in lower social welfare than a Rawlsian approach in the long-run, under reasonable conditions. We

characterize such conditions in closed-form, allowing a long-term policy comparison. In particular, the survival condition is a sufficient condition for a Rawlsian policy to achieve better social welfare in the long-run when the population of individuals satisfies homogeneous bounds on the intervention return or welfare decay.

To apply our model, the social planner does not need to know the **exact** form of $f_i, g_i$ for each individual. Rather, they can estimate general trends and effects of income shocks and interventions through small pilot experiments or through acquiring domain knowledge, e.g., through poverty trackers or longitudinal studies of intervention effects on income (Garfinkel, 2021). Experimentation through small pilot experiments is often considered a necessary precursor of policy deployment (Office, 2003; Huitema et al., 2018), rapidly increasing as a method for policy design and evaluation (Banerjee et al., 2016; 2017; Webber & Prouse, 2018). Then, the estimates for the return and decay functions can be used as plug-in estimates in the survival or the ruin condition. Future work could combine effective estimation methods for the return and decay functions with long-term policy assessments.

Our analysis rests on several modeling conditions, which could be explored in future work. We analyze several variations in the Appendix: we provide a complementary theory for the case when the survival condition is not satisfied (Appendix B), and we explore variations of our modeling assumptions: non-monotonic treatment effect function $f_i + g_i$ (Appendix D), a different tie-breaking rule (Appendix E), proportional interventions at each timestep (Appendix F), different combinations of monotonicities of return/decay functions (Appendix G), and the case of heterogenous limits of the intervention and return functions $f_i, g_i$ (Appendix H).

Overall, our theoretical framework provides versatile tools for exploring different modeling conditions as well as policy variations. These contributions open new directions for future work in the context of sequential decision-making and optimal policy design, with applications in social programs evaluation.

## 7 ACKNOWLEDGMENTS

We thank Florian E. Dorner, Jessie Finocchiaro, Max Kasy, Jon Kleinberg, Ali Shirali, Serena Wang, and anonymous reviewers for their generous feedback and detailed discussions. Rediet Abebe was partially supported by the Andrew Carnegie Fellowship Program. Jiduan Wu would like to thank the financial support from Max Planck ETH Center for Learning Systems (CLS).

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

# A    COMPLETE PROOFS

This section contains complete proofs to all results stated in the main paper. First of all, we introduce two bounds that will be repeatedly used in the following proofs:

First, consider a weighted sum of utilities:

$$\bar{U}(t) := \sum_i w_i \cdot U_i(t), \quad w_i := \left( \frac{1}{f_i^- + g_i^+} \right) \left( \sum_{j=1}^N \frac{1}{f_j^- + g_j^+} \right)^{-1}. \tag{5}$$

The increment in the weighted utility can be computed in expectation as:

$$\mathbb{E}\left[ \bar{U}(t+1) - \bar{U}(t) \mid \mathcal{F}_t \right] = \sum_i \mathbb{E}[\bar{Z}(t+1) \mid \mathcal{F}_t]$$

$$= \sum_i w_i \cdot (a_i(t) \cdot f_i(U_i(t)) - (1 - a_i(t)) \cdot g_i(U_i(t)))$$

$$\geq \sum_i w_i \cdot \left( a_i(t) \cdot f_i^- - (1 - a_i(t)) g_i^+ \right) \tag{6}$$

where $\bar{Z}(t+1) := \sum_i w_i \cdot Z_i(t+1)$. We note that the last term of equation 6 is solely a function of $\left( f_i^- \right)_i$ and $\left( g_i^+ \right)_i$. Thus, we obtain

$$\mathbb{E}\left[ \bar{U}(t+1) - \bar{U}(t) \mid \mathcal{F}_t \right] \geq \bar{\zeta}\left( (f_1^-, \dots, f_N^-), (g_1^+, \dots, g_N^+) \right). \tag{7}$$

Similarly, we define a weighted sum of utilities using slightly different weights:

$$\tilde{U}(t) := \sum_i \tilde{w}_i \cdot U_i(t), \quad \tilde{w}_i := \left( \frac{1}{f_i^+ + g_i^-} \right) \left( \sum_{j=1}^N \frac{1}{f_j^+ + g_j^-} \right)^{-1}. \tag{8}$$

Similarly, we obtain the following upper bound for $\tilde{U}(t)$:

$$\mathbb{E}\left[ \tilde{U}(t+1) - \tilde{U}(t) \mid \mathcal{F}_t \right] = \sum_{i=1}^N \tilde{w}_i \cdot (a_i(t) \cdot f_i(U_i(t)) - (1 - a_i(t)) \cdot g_i(U_i(t)))$$

$$\leq \sum_{i=1}^N \tilde{w}_i \cdot \left( a_i(t) \cdot f_i^+ - (1 - a_i(t)) g_i^- \right). \tag{9}$$

We observe that the last term of equation 9 can be written as the function $\bar{\zeta}$ with switched parameters as compared to equation 7:

$$\mathbb{E}\left[ \tilde{U}(t+1) - \tilde{U}(t) \mid \mathcal{F}_t \right] \leq \bar{\zeta}((f_1^+, \dots, f_N^+), (g_1^-, \dots, g_N^-)). \tag{10}$$

Note that both bounds from equations equation 7 and equation 10 only use the assumption on the bounds of $f_i(\cdot)$ and $g_i(\cdot)$ in Assumption 1, and hence hold under any of the aforementioned social policies and they will be crucial for the asymptotic behavior of the system.

*Proof of Theorem 3.*  We first prove the theorem for the welfare-based policy min-U, and then adapt the proof for the effect-based policy max-g.

We note that the limit conditions from Assumption 1 allow us to follow the conditions stated in Rothschild (1975b): $f_i^+ := \sup f_i(x) > 0$, $f_i^- := \inf f_i(x) > 0$, $g_i^+ := \sup g_i(x) > 0$, $g_i^- := \inf g_i(x) > 0$. Remember that $Z_i(t+1) := U_i(t+1) - U_i(t), \forall i \in [N]$. Thus, conditioning on individual $i$ getting or not getting an intervention and using the monotonicity assumptions, we obtain from the model in equation 1

$$\mathbb{E}[Z_i(t+1)|a_i(t) = 1, \mathcal{F}_t] = f_i(U_i(t)) \geq f_i^-,$$
$$-g_i^- \geq \mathbb{E}[Z_i(t+1)|a_i(t) = 0, \mathcal{F}_t] = -g_i(U_i(t)) \geq -g_i^+, \forall t. \tag{11}$$

With these conditions, together with the regularity and survival conditions, we substitute $\bar{U}(t)$ in Rothschild (1975a) with $\bar{U}(t)$ defined in equation 5 and apply Theorem 1 from Rothschild (1975a). The lower bound on equation 5 obtains the first part of the result in Theorem 1 (Rothschild, 1975a): $\liminf_{t\to\infty} U_i(t)/t \geq \bar{\zeta}$ a.s. for $i \in [N]$, which immediately implies that $\liminf_{t\to\infty} \min_i U_i(t) \to +\infty$ a.s. for $i \in [N]$.

Then, we need the following lemma.

**Lemma 1.** *Suppose $\{Y_t\}_{t\in[T]}$ are random variables and $\mathcal{F}_T$-measurable, for any $T \geq 1$. Suppose $|Y_t| \leq B$ and $\mathbb{E}[Y_t \mid \mathcal{F}_{t-1}] = \mu_t$ with $-B < \mu \leq \mu_t \leq \lambda < B$ for $\forall t \in [T]$ where $B > 0, \mu, \lambda$ are constants. Then*

$$\liminf_{T\to\infty} \frac{\sum_{t=0}^T Y_t}{T} = \liminf_{T\to\infty} \frac{\sum_{t=0}^T \mu_t}{T} \geq \mu, \quad a.s.$$

$$\limsup_{n\to\infty} \frac{\sum_{t=0}^T Y_t}{T} = \limsup_{n\to\infty} \frac{\sum_{t=0}^T \mu_t}{T} \leq \lambda, \quad a.s.$$

*Proof of Lemma 1.* The first inequality is immediate from Theorem 40 in (Freedman, 1973) with $X_t = Y_t + B$, $M_t = \mu_t + B \geq \mu + B$ and the second inequality is obtained similarly by setting $X_t = B - Y_t$, $M_t = B - \mu_t \geq B - \lambda$ instead. $\square$

Now we continue with the proof for Theorem 3. We apply Lemma 1 with $Y_t = \tilde{U}(t+1) - \tilde{U}(t)$, $B = b$, where $b$ is the upperbound on $Z_i(t+1)$ from the regularity conditions (Assumption 3), and

$$\mu_t = \mathbb{E}\left[\sum_i \tilde{w}_i(a_i(t)(f_i(U_i(t)) + g_i(U_i(t))) - g_i(U_i(t))) \mid \mathcal{F}_t\right].$$

Since $\lim_{t\to\infty} \min_i U_i(t) = \infty$, we obtain $\lim_{t\to\infty} U_i(t) = +\infty$ for all $i \in [N]$. Hence, we get $\lim_{t\to\infty} f_i(U_i(t)) = f_i^+, \lim_{t\to\infty} g_i(U_i(t)) = g_i^-$. A simple calculation finds that

$$\lim_{t\to\infty} \mu_t = \bar{\zeta}((f_1^+, \ldots, f_N^+), (g_1^-, \ldots, g_N^-)) \quad a.s. \tag{12}$$

Since $\mu_t$ is uniformly bounded for $\forall t \geq 0$, we know that $\lim_{t\to\infty} \mu_t = \lim_{T\to\infty} \frac{\sum_{t=0}^T \mu_t}{T}$, and therefore

$$\lim_{T\to\infty} \frac{\sum_{t=0}^T \mu_t}{T} = \bar{\zeta}((f_1^+, \ldots, f_N^+), (g_1^-, \ldots, g_N^-)) \quad a.s. \tag{13}$$

Hence we obtain:

$$\lim_{T\to\infty} \frac{\sum_{t=0}^T Y_t}{T} = \lim_{T\to\infty} \frac{\tilde{U}(T+1)}{T} = \bar{\zeta}((f_1^+, \ldots, f_N^+), (g_1^-, \ldots, g_N^-)) \tag{14}$$

Next, apply Lemma 1 from Rothschild (1975b) and for $\forall j \in [N]$, we have

$$\lim_{T\to\infty} \frac{\tilde{U}(T+1)}{T} = \lim_{T\to\infty} \sum_i \tilde{w}_i \cdot \left(\frac{U_i(T)}{T} - \frac{U_j(T)}{T} + \frac{U_j(T)}{T}\right) = \lim_{T\to\infty} \frac{U_j(T)}{T},$$

finalizing the proof of Theorem 3 for the welfare-based Rawlsian policy min-U. Note that Lemma 1 from Rothschild (1975b) essentially shows that the welfare gap between any two individuals converges to 0 over time, so we have used that $\lim_{T\to\infty} \sum_i \left(\frac{U_i(T)}{T} - \frac{U_j(T)}{T}\right) = 0, \forall i, j$. Intuitively, this is natural under a Rawlsian policy that always 'lifts' the lowest welfare individuals, under our bounded welfare conditions. Finally, we also used that $\sum_i \tilde{w}_i = 1$, by definition.

For the effect-based Rawlsian policy max-g, we note that if $g_i$ is strictly decreasing for all $i$, then the individual targeted at each timestep $t$ will be the exact same individual in min-U and max-g. Our modeling conditions only require that $g_i$ is decreasing, but not strictly. Therefore, if the function $g_i$ is constant for a set of individuals with welfare values under some threshold $\tau$, as long as the targeted individuals will be the one with the actual lowest welfare, min-U and max-g still coincide. Under the

tie-breaking rule of choosing the individuals with the lowest welfare, the proof for computing $R_i$'s under max-g reduces to our proof for min-U. Under different tie-breaking rules (e.g., choosing the individual with the smallest index) for max-g, the policies might actually differ in the asymptotic rates of growth. We argue that a tie-breaking rule targeting the individuals with lowest welfare under max-g policy is most natural, since it naturally applies Rawlsian principles when information gathered from $g_i$ does not help differentiate individuals.

Finally, the corollary follows immediately under the uniform boundedness assumptions on the bounds of $f_i, g_i$:

$$R_i = \bar{\zeta}((f_1^+, \ldots, f_N^+), (g_1^-, \ldots, g_N^-)) = \left( M - \sum_i \frac{g_i^-}{f_i^+ + g_i^-} \right) \cdot \left( \sum_i \frac{1}{f_i^+ + g_i^-} \right)^{-1} \tag{15}$$

$$= \left( M - N \cdot \frac{g^-}{f^+ + g^-} \right) \cdot \left( \frac{N}{f^+ + g^-} \right)^{-1} \tag{16}$$

$$= \frac{M}{N} \cdot f^+ - \frac{N - M}{N} \cdot g^- \tag{17}$$

$\square$

*Proof of Theorem 4.* We first note the intuition behind the proof, followed by the detailed technical details. We note that while max-U is also known as the 'staying with a winner' policy in Radner & Rothschild (1975), the proof technique does not generalize under non-constant functions $f_i, g_i$. To this end, we introduce a novel proof that can characterize the individual rates of growth under any informational contexts and for any functions that follow our regularity and modeling conditions (Assumptions 2(a),(b) and 3).

**Intuition:** *The main proof idea hinges on showing that a utilitarian policy tends to choose the same individuals to whom it initially allocates interventions. While the initial conditions do not change the convergence results, whoever were the first $M$ individuals to obtain an intervention at $t = 0$ have gained an advantage (a positive drift in the random process), whereas everyone else has a disadvantage (a negative drive in the random process). We bound the probability of a policy to reinforce its earlier preferred choices by the probability that an individual never drop below its initial welfare level while the other individuals never grow below their initial welfare level. Then, asymptotically, the rates of growth will converge in the following way: some fixed subpopulation converges to the maximum welfare $f^+$, whereas everyone else converges to the minimum decay $g^-$.*

First, consider the max-fg policy. Without loss of generality, consider the individual $i$ being chosen at timestep 0 ($a_i(0) = 1$). We will apply Lemma 3 for the welfare process $\{X_i(t)\}$ under an intervention, i.e., $X_i(t) = U_i(t)|a_i(t) = 1$ for all $t$, by showing that $X_i(t)$ is a submartingale and lower-bounding the probability of the welfare level decaying beyond its initial level, $X_i(0)$ (equal to the welfare initial level $U_i(0)$). This defines a random process *given that the individual $i$ will be chosen over and over again* (the process is conditioned on $a_i(t) = 1$). First, the process $\{X_i(t)\}$ is a submartingale since, conditioned on $a_i(t) = 1$,

$$\mathbb{E}[X_i(t+1)|\mathcal{F}_t] = \mathbb{E}[U_i(t+1)|\mathcal{F}_t, a_i(t) = 1] = U_i(t) + f_i(U_i(t)) > U_i(t) = X_i(t), \tag{18}$$

and note the uniform bound for $(Z_i(t))_{i,t}$ in regularity conditions (Assumption 3(c)) By an easy induction on $t$, we get that $|X_i(t)| \leq \infty, \forall t$, noting that $X_i(0) = U_i(0) < \infty$ by definition of the initial conditions.

Given our regularity conditions (Assumption 3), we may now apply Lemma 3 in a particular way: we consider $U_i(0) = u$ for some $u \in \mathbb{R}$, and we start the welfare process at $t = 1$. Note that $\{X_i(t)\}_{t \geq 1}$ is also a submartingale. Then, instead of bounding the probability of ruin, we bound the probability of $X_i(t)$ falling under the threshold $u$ (where $u := U_i(0)$). We do that by the substitution $W_t = X_i(t) - u, \forall t \geq 1$, and $W_t$ is still a submartingale. Thus, we can apply Lemma 3 to obtain

$$\psi(X_i(1)) \leq \exp(-r^* \cdot (X_i(1) - u)), \tag{19}$$

where $\psi(X_i(1)) = \mathbb{P}(T(X_i(1)) < \infty), T(X_i(1)) = \min\{t \geq 1 : X_i(t) \leq u\}$.

In a similar fashion, we now consider all other individuals $j$ who were not intervened on at the first timestep $t = 0$. For each of these, the process $\{Y_j(t)\}$, where $Y_j(t) := U_j(t)|a_j(t) = 0$ (conditioned

on $j$ *not being chosen again*) is a supermartingale:

$$\mathbb{E}[Y_i(t+1)|\mathcal{F}_t] = \mathbb{E}[U_i(t+1)|\mathcal{F}_t, a_i(t) = 0] = U_i(t) - g_i(U_i(t)) < U_i(t) = Y_i(t) \qquad (20)$$

by applying equation 1 and noting the all functions $g_i(\cdot)$ are positive by definition. In addition, again we know $|Y_i(t)| < \infty$ by uniform boundedness of $|Z_i(t)|$ in regularity conditions(Assumption 3(b)), and noting that $Y_i(0) = U_i(0) < \infty$ by definition of the initial conditions.

Again, we can apply Lemma 3 for the process $\{-Y_j(t)\}_{t \geq 1}$ (which is now a submartingale) and ruin threshold $-u$ to obtain

$$\psi(-Y_j(1)) \leq \exp(-r^* \cdot (u - Y_j(1))), \qquad (21)$$

where $\psi(-Y_j(1)) = \mathbb{P}(T(Y_j(1)) < \infty), T(Y_j(1)) = \min\{t \geq 1 : -Y_j(t) \leq -u\}$.

Next, we lower bound the probabability that the individuals who were chosen at timestep $0$, denoted as set $S$, will continue to be chosen at every timestep. To do so, we note that this probability is equal to the probability that every $i \in S$ is chosen at every subsequent $t \geq 1$ *and* all other $j \notin S$ are not chosen at every $t \geq 1$. Among all events that comprise this probability, one of them is the event in which $X_i(t) \geq u$ and $X_j(t) \leq u, \forall j \notin S$ (remember here that $X_i(t) = U_i(t)|a_i(t) = 1$ and $Y_j(t) = U_j(t)|a_j(t) = 0$, for $j \notin S$). Thus,

$$\mathbb{P}(a_i(t) = 1 \text{ and } a_j(t) = 0, \forall j \notin S, \forall t \geq 1) \geq \mathbb{P}(X_i(t) \geq u \text{ and } Y_j(t) \leq u, \forall j \notin S, \forall t \geq 1) \quad (22)$$

The righthandside consists of independent events w.r.t. $t$, since we have conditioned already on the intervention, so we can further compute it as

$$\prod_{i \in S} \mathbb{P}(X_i(t) > u, \forall t \geq 1) \cdot \prod_{j \notin S} \mathbb{P}(Y_j(t) < u, \forall t \geq 1)$$
$$= \prod_{i \in S} (1 - \mathbb{P}(\exists T < \infty \text{ s.t. } X_i(T) \leq u)) \cdot \prod_{j \notin S} (1 - \mathbb{P}(\exists T < \infty \text{ s.t. } Y_j(T) \geq u)) \qquad (23)$$

Finally, we lowerbound equation 23 by the bound we obtained by our Lundberg-type inequality for the welfare process:

$$\prod_{i \in S} (1 - \mathbb{P}(\exists T < \infty \text{ s.t. } X_i(T) \leq u)) \cdot \prod_{j \notin S} (1 - \mathbb{P}(\exists T < \infty \text{ s.t. } Y_j(T) \geq u)) \qquad (24)$$

$$\geq \prod_{i \in S} (1 - \psi(X_i(1))) \cdot \prod_{j \notin S} (1 - \psi(-Y_j(1))) \qquad (25)$$

$$\geq \prod_{i \in S} (1 - \exp(-r^* \cdot (X_i(1) - u))) \cdot \prod_{j \notin S} (1 - \exp(-r^* \cdot (u - Y_j(1)))) \qquad (26)$$

Our regularity conditions ensure that equation 26 is lowerbounded by some positive constant $p_i^* > 0$: Assumption 3 states that $\exists z^*, l > 0$ s.t. $\mathbb{P}(Z_i(t+1) \geq z^*|\mathcal{F}_t) \geq l$ and $\mathbb{P}(Z_i(t+1) \leq -z^*|\mathcal{F}_t) \geq l$. Since $l > 0$, this offers a strictly positive lower bound on equation 26. Furthermore, $p_i^*$ does not depend on $\mathcal{F}_0$ but it may depend on the initial individual $i$ that was intervened on at timestep $0$. We take the minimum of $p_i^*$ among all individuals $i \in [N]$ (since any of them could have been intervened on at timestep $0$), and obtain $p^* := \min\{p_i^*\} > 0$. Then, note that the probability of individual $i$ being chosen for all $t \geq 0$ also depends the rule of max-fg and the tie-breaking rule of choosing the smallest index, which will ensure the individual being constantly chosen once the $f_i(U_i(t)) + g_i(U_i(t)) \geq f_j(U_j(t)) + g_j(U_j(t))$ won't be violated for all $t \geq 0$. This is true since in addition to the modeling condition that states that $g_i(\cdot)$ is decreasing, we also assumed that $f_i(\cdot) + g_i(\cdot)$ is increasing.

As time grows, the probability of the utilitarian policy fixating on one single individual is lowerbounded by $1 - (1 - p^*)^m$ where $m$ denotes the number of times the set of individuals who receive the intervention changes, which converges to $1$ as $m \to \infty$.

Lastly, we prove that for an individual $i \in S$ with $a_i(t) = 1$ for all $t \geq 0$ a.s., we have $R_i = f^+$ a.s., and for an individual $j$ with $a_j(t) = 0$ for all $t \geq 0$ a.s., $j \notin S$, we have $R_j = -g^+$ a.s.

In doing so, we apply Lemma 1 repeatedly:

- First, for individual $i \in S$ that gets chosen at the first timestep, we set $Y_i = U_i(t+1) - U_i(t)$, $\mu = f^-$ and obtain from Lemma 1 that $\lim_{t \to \infty} U_i(t) = +\infty$. For the same individual, we can apply Lemma 1 again with $\mu = \lambda = f^+$, which shows convergence of $(U_i(t) - U_i(0))/t$ to $f^+$. We then note that $\lim_{t \to \infty}(U_i(t) - U_i(0))/t = R_i = f^+$.

- Second, for any individual $j \notin S$, we set $Y_t = U_j(t+1) - U_j(t)$ and $\lambda = -g^-$ and obtain from Lemma 1 that $\lim_{t \to \infty} U_j(t) = -\infty$. Finally, we apply Lemma 1 again for all such $j$ with $\mu = \lambda = -g^+$, which shows convergence of $(U_j(t) - U_j(0))/t$ to $-g^+$. We then note that $\lim_{t \to \infty}(U_j(t) - U_j(0))/t = R_j = -g^+$.

We note that the proof goes through in the exact same way for the max-U and max-f policies, since the only place the functions $f_i$ and $g_i$ play a role is in the tie-breaking rule: when $f_i$ is increasing and the tie-breaking rule always chooses the individual with the lowest index, the probability of a policy continuing to choose the same set of individuals converges to 1, whereas the probability of every choosing another individual $j$ converges to 0. □

*Proof of Theorem 5.* By the weak homogeneity condition (Assumption 3(c)) and the survival condition (Assumption 1), we have $f^- > \frac{N-M}{M}g^+$. Then, we apply Lemma 1 by setting $Y_t = U_i(t+1) - U_i(t)$ and $\mu_t = \frac{M}{N}f_i(U_i(t)) - (1 - \frac{M}{N})g_i(U_i(t))$. We note that $\mu_t$ actually evaluates in expectation the rate of welfare increase under the random policy, where $\mathbb{E}[a_i(t) \mid \mathcal{F}_t] = \frac{M}{N}$ for any $t \geq 0$, $i \in [N]$ and $\mathcal{F}_t$. Thus, we obtain that $\mathbb{E}[U_i(t+1) - U_i(t) \mid \mathcal{F}_t] \geq \frac{M}{N}f_i^- - (1 - \frac{M}{N})g_i^+ > 0$ under the random policy. From this we conclude that $\lim_{t \to \infty} \min_i U_i(t) = \infty$, and thus every individual's welfare will increase unboundedly over time. Since $\lim_{T \to \infty} U_i(T) \to +\infty$, we have $\lim_{T \to \infty} \mu_T = \frac{M}{N}f^+ - \left(1 - \frac{M}{N}\right)g^-$ and since $\mu_T$ is bounded, we apply Lemma 1 with $Y_t = U_i(t+1) - U_i(t)$, $\lambda = \mu = \frac{M}{N}f^- - (1 - \frac{M}{N})g^+$. Finally we conclude

$$R_i = \lim_{t \to \infty} \frac{U_i(t) - U_i(0)}{t} = \frac{M}{N}f^- - \left(1 - \frac{M}{N}\right)g^+ > 0, \quad a.s. \tag{27}$$

□

## A.1 LUNDBERG'S INEQUALITY FOR SUBMARTINGALES

In this subsection, we present technical details used in the proof of Theorem 4.

In Moriconi (1986) (page 179), the author briefly mentioned the Lundberg's inequality also holds for submartingales. We provide the proof below for completeness. First, we define the adjustment coefficient for submartingales:

**Definition 2.** *Let $\{X_t\}$ be a submartingale, the adjustment coefficient, denoted by $r^*$, is the positive value such that $\{\exp(-r^* X_t)\}$ is a martingale, i.e., $\mathbb{E}[\exp(-r^* Z_{t+1})] = 1$ where $Z_{t+1} := X_{t+1} - X_t$.*

**Lemma 2.** *(Lundberg's inequality for submartingales) Let $\{X_t\}$ be a submartingale with $X_0 = u > 0$, $r^*$ be the adjustment coefficient of $\{X_t\}$ and assume $\{Z_t\}$ are i.i.d.. The probability of ultimate ruin is bounded as follows*

$$\psi(u) \leq \exp(-r^* u)$$

*where $\psi(u) := \mathbb{P}(T(u) < \infty)$, $T(u) := \min\{t \geq 1 : X_t \leq 0, X_0 = u\}$.*

*Proof.* The proof is similar to the one for analyzing the surplus of an insurance portfolio (refer to Theorem 5.2 in Tse (2023)). We prove the result by induction on $t$, for $\psi(t; u) := \mathbb{P}(T(u) \leq t)$ and

denoting

$$\psi(1; u) = \int_{-\infty}^{-u} \mathbb{P}(Z_2 = x) dx$$

$$\leq \int_{-\infty}^{-u} \exp(r^*(-u - Z_2)) \mathbb{P}(Z_2 = x) dx$$

$$= \exp(-r^* u) \cdot \int_{-\infty}^{-u} \exp(-r^* Z_2) \mathbb{P}(Z_2 = x) dx$$

$$\stackrel{(a)}{=} \exp(-r^* u).$$

Assume Lundberg inequality holds for any time step less than $t$ and $u > 0$ and now consider $\psi(t + 1, u)$,

$$\psi(t + 1; u) = \psi(1, u) + \int_0^\infty \psi(t, x) \mathbb{P}(Z_2 = x - u) dx$$

$$\stackrel{(b)}{\leq} \int_{-\infty}^{-u} \mathbb{P}(Z_2 = x) dx + \int_0^\infty \exp(-r^* x) \mathbb{P}(Z_2 = x - u) dx$$

$$\leq \int_{-\infty}^{-u} \exp(-r^*(x + u)) \mathbb{P}(Z_2 = x) dx + \int_{-u}^\infty \exp(-r^*(x + u)) \mathbb{P}(Z_2 = x) dx$$

$$= \exp(-r^* u) \mathbb{E}[\exp(-r^* Z_2)] \stackrel{(c)}{=} \exp(-r^* u).$$

where inequality (b) holds by Lundberg's inequality for time step $t$. □

**Remark 1.** *Note in the proof of Lemma 2, using condition $\mathbb{E}[\exp(-r^* Z_2)] \leq 1$ in equality (a),(c) is enough, which is a weaker condition than $Z_2$ is adjustable.*

The following corollary is an immediate result of the above lemma.

**Corollary 3.** *Let $\{X_t\}$ be a submartingale with $\mathbb{E}[X_{t+1} \mid X_t] = X_t + c$ where $c > 0$. Denote $Z_{t+1} := X_{t+1} - X_t$ and $\{Z_t\}$ are i.i.d. Assume there $\exists z^* > 0$ s.t. $\mathbb{P}(Z_{t+1} \geq z^*) > 0$ and $\mathbb{P}(Z_{t+1} \leq -z^*) > 0$. There exists a positive constant $r^*$ such that*

$$\psi(u) \leq \exp(-r^* u).$$

*Proof.* The proof is immediate by noticing

$$\mathbb{E}[\exp(-r X_{t+1}) \mid \mathcal{F}_t] = \exp(-r X_t) \cdot \mathbb{E}[\exp(-r(X_{t+1} - X_t)) \mid \mathcal{F}_t].$$

Denote $\phi(r) := \mathbb{E}[\exp(-r^*(X_{t+1} - X_t)) \mid \mathcal{F}_t]$, which is continuously differentiable, and we obtain $\phi(0) = 1, \phi'(0) = -c$ by computing the closed-form derivative. Since $\mathbb{P}(Z_{t+1} \geq z^*) > 0$, $\mathbb{P}(Z_{t+1} \leq -z^*) > 0$, and hence we have $\lim_{r \to +\infty} \phi(r) = +\infty$. Hence there exists $r^* > 0$ such that $\phi(r^*) = 1$. Moreover, since $Z_t$ are i.i.d. and $\{X_t\}$ is adjustable (i.e., there exists an adjustment coefficient as defined in Definition 2 that does not depend on $\{X_t\}$), we can apply Lemma 2 and we conclude the proof. □

The following lemma is an adapted version of Lemma 2 and will become useful in the proof of the main result.

**Lemma 3.** *(Lundberg's inequality for a welfare process) Consider a random process $\{X_i(t)\}$ defined as $X_i(t) = U_i(t) | a_i(t) = 1$, with $U_i(0) = u$ and $\{U_i(t)\}$ defined as the welfare process in model 1, for $i = 1, \ldots, N$. As such, $\{X_i(t)\}$ defines a welfare process under an intervention, i.e. $a_i(t) \equiv 1$. Assume there exists $z^* > 0$, $0 < l < 1$ s.t. $\mathbb{P}(Z_i(t + 1) \geq z^* \mid \mathcal{F}_t) \geq l$, $\mathbb{P}(Z_i(t + 1) \leq -z^*) \geq l$ for any $\mathcal{F}_t$ and any $i$. Then, for an individual $i$, there exists a positive constant $r^*$, such that the probability of ultimate ruin is bounded as follows*

$$\psi(u) \leq \exp(-r^* u) \tag{28}$$

*where, by an abuse of notation, $\psi(u) := \mathbb{P}(T(u) < \infty)$, $T(u) := \min\{t \geq 1 : X_i(t) \leq 0, X_i(0) = u\}$.*

*Proof.* For an individual $i$ and a timestep $t$, denote $\phi(r) := \mathbb{E}[\exp(-r(X_i(t+1) - X_i(t))) \mid \mathcal{F}_t]$, for any $r$. We observe

$$\phi(r) = \sum_{k=-b}^{b} \mathbb{P}(Z_i(t+1) = k) \exp(-rk),$$

$$\phi'(r) = \sum_{k=-b}^{b} -k\mathbb{P}(Z_i(t+1) = k) \exp(-rk),$$

$$\phi''(r) = \sum_{k=-b}^{b} k^2 \mathbb{P}(Z_i(t+1) = k) \exp(-rk).$$

Notice that $\phi(0) = 1$, $\phi'(0) = -\mathbb{E}[Z_i(t+1) \mid \mathcal{F}_t] \leq -g^-$, and $0 < \phi''(r) \leq 2bk^2 \exp(rb)$ for any $\mathcal{F}_t$. First of all, we know that there exists a small interval for $r$ near zero, $(0, a]$ for some $r(\mathcal{F}_t) > 0$, such that $\phi(x) \leq 1$ for $x \in (0, r(\mathcal{F}_t)]$ by applying Rolle's theorem. Next, we claim there exists $r^* > 0$ s.t. $\phi(r^*) \leq 1$, that is independent of any $\mathcal{F}_t$. This claim can be easily proved by contradiction: assume for any $\epsilon > 0$, there exists $r_\epsilon \leq \epsilon$, $\mathcal{F}_t(\epsilon)$ such that $\phi(r_\epsilon) > 1$. Note that under $\mathcal{F}_t(\epsilon)$ and since $\phi(0) = 1$, there must be an $x \in (0, r_\epsilon)$ such that $\phi'(r_\epsilon) > 0$. Making $\epsilon$ arbitrarily small, we get that $r_\epsilon \to 0$ and since $\phi(0) = 1$ and $\phi'(0) < 0$, there must exist $x \in (0, r_\epsilon]$ such that $\phi''(x) \to +\infty$. This contradicts with the fact that $\phi''(r) \leq 2bk^2 \exp(rb)$ for any $\mathcal{F}_t$. Hence for our welfare process $Z_i(t)$, which is not i.i.d. for different $t$, and not adjustable, but it satisfies $\mathbb{E}[\exp(-r^* Z_i(t))] \leq 1$ for some positive constant $r^*$ that is independent of $\mathcal{F}_t$ and all $i \in [N]$.

Finally, we conclude the proof by pointing out that the proof of Lemma 2 still applies under condition $\mathbb{E}[\exp(-r^* Z_i(t))] \leq 1$ (see Remark 1). □

## B  POLICY COMPARISON UNDER A RUIN CONDITION

Our survival condition, Assumption 1, defined a parameter condition in which there exists a policy that can 'lift' every individual unboundedly, as time grows. We show that different versions of the Rawlsian policy achieve this property (in addition, the constant proportions policy from Radner & Rothschild (1975) will also achieve this property). Under the survival condition, our main result shows that the Rawlsian policy will achieve better long-term social welfare.

In this section, we introduce a complementary condition to the survival condition, called a *ruin condition*. Intuitively, under this condition, even a Rawlsian policy will not be able to ensure that every individual will have positive welfare, asymptotically. As such, the lowest welfare will decay indefinitely almost surely.

*Note: we borrow the 'ruin' terminology from ruin theory, but the definition of a 'ruin condition' is specific to our setting, as defined below. Our proofs make use of ruin theory in applying Lundberg's inequality, as seen in Appendix A.*

**Assumption 4** (Ruin condition). *We assume $\bar{\zeta}((f_1^+, \ldots, f_N^+), (g_1^-, \ldots, g_N^-)) < 0$ where $\bar{\zeta} : \mathbb{R}^{2N} \to \mathbb{R}$ is defined in equation 3. and $f_i^+ := \sup f_i(x) > 0$, $f_i^- := \inf f_i(x) > 0$, $g_i^+ := \sup g_i(x) > 0$, $g_i^- := \inf g_i(x) > 0$.*

**Theorem 6** (Theorem 2, formal). *If the ruin condition is met, under regularity, modeling conditions, and as long as $f_i(\cdot) + g_i(\cdot)$ is increasing for all $i \in [N]$, the result in Theorem 1 is reversed:*

$$\bar{R}_{\text{utilitarian}} \geq \bar{R}_{\text{Rawlsian}} \quad a.s.$$

*where the Rawlsian and utilitarian policies are defined in the same informational contexts, i.e.* $(\text{min-U}, \text{max-U})$, $(\text{max-g}, \text{max-f})$, $(\text{max-g}, \text{max-fg})$.

*Proof of Theorem 6.* We prove this result similarly as in Theorem 1, by computing in closed-form the individual rates of growth under every policies, and then computing the long-term social welfare as an average of these rates. First, we note that we can compute the individual rates of growth under utilitarian policies just like in Theorem 4 by noting that the proof does not make use of the survival condition (the survival condition is only necessary to compute the individual rates of growth under Rawlsian policies). Next, we compute the individual rates of growth for Rawlsian policies

under the ruin condition in Theorem 7 (Corollary 4). We then use uniform boundedness to compute the long-term social welfare for Rawlsian and utilitarian policies, noting that a utilitarian policy is better than a Rawlsian policy as long as $f^+ \geq f^-$, which is true by the "rich-get-richer" modeling condition. □

We present a visualization of Theorem 6 in Figure 2 where we can observe the utilitarian policies (max-U, max-f, max-fg) converge to a higher growth rate while Rawlsian policies (min-U, max-g) converges to a suboptimal average growth rate. The experiment setting here is as same as Section 5.1. Under uniform boundedness, the parameters for the shape of $(f_i(\cdot))_i$, $(g_i(t))_i$ are randomly sampled within the same interval, which is weaker than the assumption in Theorem 6.

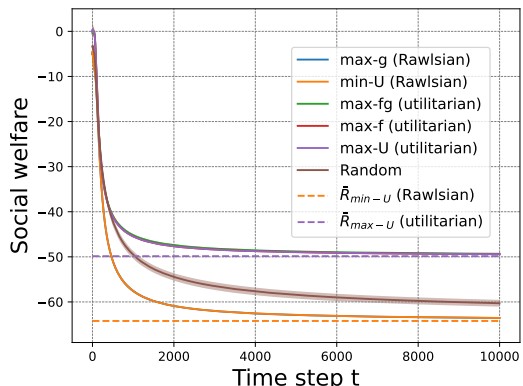

Figure 2: We show social welfare as the finite-time growth rate averaged over all individuals, for all policies (solid lines), as well as theoretical expected growth rate (dashed lines) under the ruin condition.

**Theorem 7.** *Under modeling, regularity, and ruin conditions (Assumption 2.(a),(b), 3, 4), the* min-U *policy leads to the following closed form solution of the individual rates of growth:*

$$R_i = \bar{\zeta}((f_1^-, \dots, f_N^-), (g_1^+, \dots, g_N^+)), \quad i = 1, \cdots N, \quad a.s.$$

**Corollary 4.** *With the addition of the uniform boundedness condition from Assumption 2.(c), we can simplify the individual rates of growth, obtaining the long-term social welfare value for the Rawlsian policy under the ruin condition*

$$\bar{R}_{\min-\mathrm{U}} = \bar{R}_{\max-\mathrm{g}} = \frac{M}{N}f^- - \frac{N-M}{N}g^+ < 0 \quad a.s.$$

*Proof of Theorem 7.* Under the ruin condition, consider the upper bound in inequality 10 and we obtain

$$\mathbb{E}[\tilde{U}(t+1) - \tilde{U}(t) \mid \mathcal{F}_t] \leq \bar{\zeta}((f_1^+, \dots, f_N^+), (g_1^-, \dots, g_N^-)) < 0.$$

Applying Lemma 1 with $Y_t = \tilde{U}(t+1) - \tilde{U}(t)$, $\mu = \bar{\zeta}((f_1^+, \dots, f_N^+), (g_1^-, \dots, g_N^-))$, we obtain $\tilde{U}(t+1) \to -\infty$ a.s. and hence $\min_{i \in [N]} U_i(t) \to -\infty$. Here we use the same intuition as Rothschild (1975a), proving there exists $T^*$ such that

$$\lim_{t \to \infty} \frac{\max_i U_i(T^*) - \min_j U_j(T^*)}{t} = 0. \tag{29}$$

We prove equation 29 by applying Proposition 1 and Lemma 1 in Rothschild (1975a). Now with equation 29, we know everyone has the same growth rate and hence $U_i(t) \to -\infty$ a.s. for all $i \in [N]$. Then by the uniform boundedness from our modeling condition and the monotonicity of $(f_i(\cdot))_i$, $(g_i(\cdot))_i$, we have

$$\lim_{t \to \infty} \mathbb{E}\left[\frac{\sum_{i=1}^{N}(U_i(t+1) - U_i(t))}{Nt}\right] = \frac{M}{N}f^- - \frac{N-M}{N}g^+$$

Apply Lemma 1 with $Y_t = \frac{\sum_{i=1}^{N}(U_i(t+1)-U_i(t))}{N}$, $\lambda = \mu = \frac{M}{N}f^- - \frac{N-M}{N}g^+$, we conclude

$$R_i = \frac{M}{N}f^- - \frac{N-M}{N}g^+, i = 1, \cdots, N.$$

Then since every individual has the same growth rate, we conclude our proof and our corollary follows immediately. We note that the same result easily follows for the max-g policy, given the tie-breaking rule that favors the individual with the lowest welfare. $\square$

**Remark 2.** *To show convergence of the individual rates of growth under the ruin condition, uniform boundedness is needed to obtain the same growth rate for every individual. In contrast, Theorem 3 does not require uniform boundedness for obtaining the same individual growth rate, asymptotically.*

The following proposition is a counterpart for Proposition 2 in Rothschild (1975a) under the ruin conditions. In the proof below, we emphasize the differences while keeping the other steps concise.

**Proposition 1.** *Under the conditions of Theorem 7, let $D(t) := \max_{i \in [N]} U_i(t) - \min_{j \in [N]} U_j(t)$, there exists a constant $G$ such that if $D(s) \geq G$ and $T^*$ is the first integer such that $D(s+T^*) < G$, then there exist $H$ and $K$ such that $\mathbb{P}(T^* > n) \leq He^{-nK}$.*

*Proof of Proposition 1.* Suppose $\mathcal{K}$ is any proper subset of $[N]$, and $\mathcal{K}'$ is the complement of $\mathcal{K}$ in $[N]$. We take the case where the min-U policy only considers individuals in $\mathcal{K}$ while ignoring individuals in $\mathcal{K}'$. We prove the following inequality by induction: there exists a constant $T_{\mathcal{K}}$ such that

$$\mathbb{E}[\max_{j \in \mathcal{K}'} U_j(t) - \min_{i \in \mathcal{K}} U_i(t) \mid \mathcal{F}_0] \leq \max_{j \in \mathcal{K}'} U_j(0) - \min_{i \in \mathcal{K}} U_i(0) - 2, \quad \forall t \geq T_{\mathcal{K}}. \quad (30)$$

The inequality in equation 30 allows us to satisfy the conditions in Proposition 2 from Rothschild (1975a) and easily adapt the proof of Theorem 7. The base case $N = 2$ is trivial since we know everything about the behavior of an individual $i$ with $a_i(t) = 0$ for all $t \geq 0$. Now assume that for $N = N_0$, Proposition 1 holds, and consider $N = N_0 + 1$. Consider the set $\mathcal{K}$, since $|\mathcal{K}| \leq N_0$ and by induction, we have

$$\lim_{t \to \infty} \frac{U_i(t)}{t} = \bar{\zeta}((f_i^-)_{i \in \mathcal{K}}, (g_i^+)_{i \in \mathcal{K}}), \quad \forall i \in \mathcal{K}, \quad a.s.$$

where $\bar{\zeta}_{\mathcal{K}}((x_i)_{i \in \mathcal{K}}, (y_j)_{j \in \mathcal{K}}) := \left(M - \sum_{k \in \mathcal{K}} \frac{y_i}{x_i+y_i}\right)\left(\sum_{i \in \mathcal{K}} \frac{1}{x_i+y_i}\right)^{-1}$. As for the set $\mathcal{K}'$, apply the monotonicity of $(g_i(\cdot))_{i \in \mathcal{K}'}$ and the uniform boundedness condition, obtaining

$$\lim_{t \to \infty} \frac{U_i(t+1)}{t} = -g^+, \quad \forall i \in \mathcal{K}', \quad a.s.$$

Hence we know that

$$\lim_{t \to \infty} \frac{\max_{j \in \mathcal{K}'} U_j(t) - \min_{i \in \mathcal{K}} U_i(t)}{t} = -g^+ - \bar{\zeta}((f_i^-)_{i \in \mathcal{K}}, (g_i^+)_{i \in \mathcal{K}}) < 0, a.s.$$

By applying the Fatou-Lebesque theorem, we have

$$\lim_{t \to \infty} \mathbb{E}\left[\frac{\max_{j \in \mathcal{K}'} U_j(t) - \min_{i \in \mathcal{K}} U_i(t)}{t} \mid \mathcal{F}_0\right] = -g^+ - \bar{\zeta}((f_i^-)_{i \in \mathcal{K}}, (g_i^+)_{i \in \mathcal{K}}) < 0,$$

Then there exists $T_{\mathcal{K}} > 0$ such that for all $t > T_{\mathcal{K}}$ and

$$\mathbb{E}\left[\max_{j \in \mathcal{K}'} U_j(t) - \min_{i \in \mathcal{K}} U_i(t) \mid \mathcal{F}_0\right] \leq \max_{j \in \mathcal{K}'} U_j(0) - \min_{i \in \mathcal{K}} U_i(0) - 2.$$

At this point, we may apply Lemma 4 and Lemma 5 from Rothschild (1975a) and conclude our proof. $\square$

**Remark 3.** *The intuition for Proposition 1 is the following: when the ruin condition holds, individuals receiving an allocation will still decay in welfare, due to the strong decay functions effects that the ruin condition models. However, this happens at a slower rate compared to individuals whose welfare decays* absent *any intervention. As such, the welfare gap between the individuals with the maximum welfare level and those with minimum welfare will be bounded, asymptotically, and therefore everyone will decay, yet at a slower rate given the intervention of the social planner than without any intervention.*

**Remark 4.** *The survival and ruin conditions characterize two model states in which we can make a definite comparison between Rawlsian policies and utilitarian policies in terms of the long-term social welfare they achieve. There is a middle ground, in which neither survival nor ruin may hold, in which the direct comparison between policies becomes much more difficult. We leave this direction for future studies.*

## C  EXPERIMENTAL DETAILS

This section contains detailed simulation notes for Section 5 and Appendix figures. For the simulations in which the return and decay function bounds are uniform, we choose threshold parameters $F_i^-, F_i^+, G_i^-, G_i^+$ s.t. $f_i(x) = f^-, g_i(x) = g^+$ for $x \leq F_i^-, x \leq G_i^-$, respectively, and $f_i(x) = f^+$, $g_i(x) = g^-$ for $x \geq F_i^+, x \geq G_i^+$, respectively, and we linearly interpolate between these thresholds. Choosing $F_i^- < F_i^+$ and $G_i^- > G_i^+$ ensures that $f_i$ is increasing and $g_i$ is decreasing on the non-constant segments. We generate $F_i^-, F_i^+, G_i^-, G_i^+$ randomly in the interval $(0, \Delta]$ for some $\Delta > 0$. For Figures 1, 2, and 4, we filter to ensure that $f_i(\cdot) + g_i(\cdot)$ is increasing. For Figure 3, we filter to ensure that $f_i(\cdot) + g_i(\cdot)$ is increasing under some threshold $\tau$, and decreasing above threshold $\tau$. For all figures, we average over 50 iterations and report the social welfare obtained at every timestep. Our results are qualitatively the same for other functional forms of $f_i, g_i$, such as sigmoid functions.

All code and data used in our simulations is available in this repository.

## D  BEYOND A MATTHEW EFFECT: MODELING VARIATIONS OF THE TREATMENT EFFECT FUNCTION

In the main text, we modeled a Matthew effect through the "rich-get-richer" and "poor-get-poorer" behaviors induced by an increasing intervention return function $f_i(\cdot)$ and a decreasing decay function $g_i(\cdot)$, under the assumption that the treatment effect $f_i(\cdot) + g_i(\cdot)$ is also increasing. This assumption suggests that interventions at higher level of welfare have a higher impact. We explore a variation of this assumption in this section, assuming that there exists a threshold $\tau$ above which the treatment effect is in fact decreasing. In doing so, we capture a diminishing return effect, where individuals with the highest or lowest levels of welfare benefit less from an intervention than individuals with moderate levels of welfare. This is motivated by recent policies that target people with moderate welfare values: the algorithmic profiling policy introduced by Austria in 2020 (Allhutter et al., 2020) predicts a probability of an individual to re-enter the job market based on an intervention (in a sense, a prediction of the treatment effect). The policy allocates an intervention to those "in-the-middle", suggesting that moderate welfare values are predictive of the highest treatment effect. In addition, optimal taxation policy and redistributive taxation (Mankiw et al., 2009; Benabou, 2000) often argue for an increasing tax scale or a decreasing benefit scheme as a function of income. We provide a theoretical extension from our results in Theorem 1 under stricter homogeneity assumptions, capturing a diminishing return on interventions.

**Corollary 5** (Diminishing returns)**.** *Assume the conditions of Theorem 1, but with a threshold $\tau > 0$ s.t. $f_i(x), f_i(x) + g_i(x)$ are decreasing for $x \geq \tau$. Furthermore, assume that the functions $f_i, g_i$ are uniform for all individuals ($f_i(x) = f_j(x), g_i(x) = g_j(x)$ for $\forall i \neq j \in [N]$). Finally, $\lim_{x \to -\infty}(f_i(x) + g_i(x)) < \lim_{x \to +\infty}(f_i(x) + g_i(x))$. Then with positive probability, a Rawlsian policy achieves a higher long-term social welfare than a utilitarian policy:*

$$\mathbb{P}(\bar{R}_{\text{Rawlsian}} \geq \bar{R}_{\text{utilitarian}}) > 0,$$

*where the Rawlsian and utilitarian policies are defined in the same informational contexts, i.e. $\{\text{min-U}, \text{max-U}\}, \{\text{max-g}, \text{max-f}\}, \{\text{max-g}, \text{max-fg}\}$. Note here that all policies break the tie by choosing the individual with the lowest welfare level.*

**Remark 5.** *Corollary 5 shows that our analysis under the simple assumptions on the monotonicies of intervention return function and the decay function can be applied to more complicated cases. When $f_i(\cdot) + g_i(\cdot)$ is increasing, the choices of max-fg (max-f) policy and min-U policy diverge. The tendency of max-fg (max-f) policy of focusing on the better-off population can cause long-term loss by accumulating the decay of the ignored population. Furthermore, the ignored population enters a low-welfare trap, since they will likely not be targeted again. See Figure 3 for an illustration*

*of Corollary 5 where we can observe utilitarian policies (*max-U, max-f, max-fg*) show a lower growth rate over the finite time horizon as compared to the Rawlsian policies (*min-U, max-g*).*

*Proof.* First of all, since the asymptotic behavior of individuals under the min-U policy (i.e. being lifted unboundedly) does not depend on the simple monotonicity property of the return functions, we have $U_i(t) \to +\infty$ for $\forall i \in [N]$ under a Rawlsian policy with survival, regularity conditions and existence of $\lim_{x \to -\infty}(f_i(x) + g_i(x))$ from Theorem 3. Hence we conclude $\bar{R}_{\text{Rawlsian}} = \frac{M}{N}f^+ - \frac{N-M}{N}g^-$. However, before the turning point of the monotonicity of $f_i(\cdot) + g_i(\cdot)$, the max-fg policy tends to focus on the better-off individuals by applying the rule of max-fg policy. Hence with positive probability, some individuals will be left behind (with $f_i(U_i(t)) + g_i(U_i(t))$ less than $\lim_{x \to +\infty}(f_i(x) + g_i(x)))$; then if these individuals will not receive budget for all $t$ afterwards, the probability of them never crossing the turning point (i.e., where the mononicity of the treatment effect function changes) is lowerbounded by a positive constant independent of $\mathcal{F}_t$ by applying Lemma 3.

And after the turning point, the max-fg policy coincides with the min-U policy and lifts every individual to infinity, asymptotically, with positive probability (not almost surely anymore since the individuals can drop below the turning point of $f_i(\cdot) + g_i(\cdot)$). Hence we conclude that with positive probability, $\bar{R}_{\text{max-fg}} \in \left\{ \frac{Mf^+ - (k-1)g^- - (N-M-k)g^+}{N}, k = 1, \ldots, N - M. \right\}$. With positive probability, we have $\bar{R}_{\text{max-fg}} \leq \bar{R}_{\text{min-U}}$.

A similar argument applies to the max-f policy by substituting $f_i(\cdot) + g_i(\cdot)$ with $f_i(\cdot)$ and hence omitted here. For the max-U policy, Theorem 4 still applies and we have $\bar{R}_{\text{max-fg}} \leq \bar{R}_{\text{min-U}}$ with positive probability. □

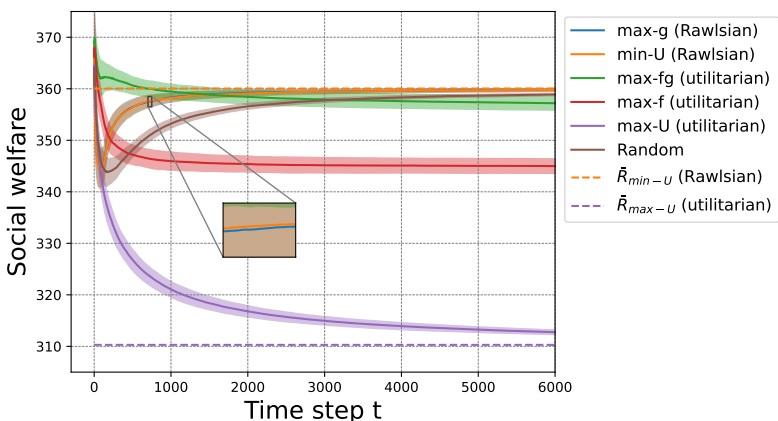

Figure 3: We show social welfare as the finite-time growth rate averaged over all individuals, for all policies (solid lines), as well as theoretical growth rate for min-U and max-U policies (dashed lines) under diminishing returns.

# E    DIFFERENT TIE-BREAKING RULE FOR THE MAX-G POLICY

In Section 2 we introduced a rule that breaks the tie in favor of individuals with the smallest index, when they have the same welfare values. Additionally, when the decay function values are the same, the max-g policy chooses the individual with the lowest welfare. We explore a variation where the max-g policy breaks the tie by also choosing the individual with the lowest welfare in Figure 4, noting a slightly convergence rate than the min-U policy. All simulations details are the same as in Section 5.

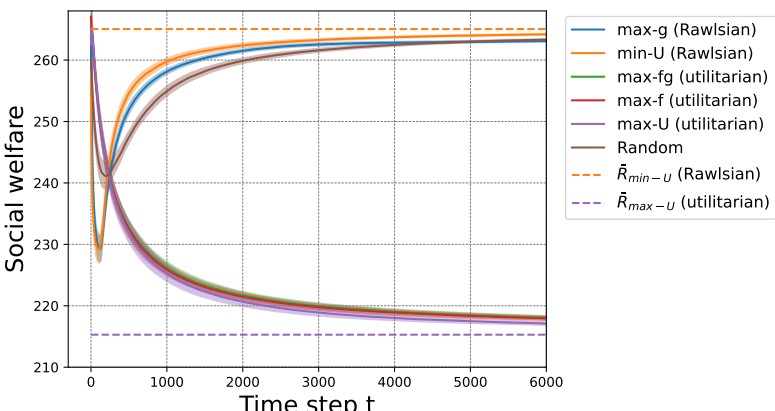

Figure 4: Social welfare as the finite-time growth rate averaged over all individuals, for all policies (solid lines), as well as theoretical expected growth rate, asymptotically (dashed lines). The tie-breaking rule favors the individual with the lowest index for all policies.

## F  PROPORTIONAL RESOURCE ALLOCATION

In the main text of the paper, we restrict our attention to integer resources at each time step and use a budget $\geq 1$ to intervene in several individuals. Here, we consider interventions that allow proportional allocation, as follows:

$$\text{proportional max-U: } a_i(t) = \frac{e^{U_i(t)}}{\sum_{j=1}^{N} e^{U_j(t)}},$$

$$\text{proportional min-U: } a_i(t) = \frac{e^{-U_i(t)}}{\sum_{j=1}^{N} e^{-U_j(t)}}.$$

**Theorem 8.** *Under regularity (Assumption 3) and modeling conditions (Assumption 2), and assume that $f_i(x) \equiv f(x)$, $g_i(x) \equiv g(x)$, the proportional max-U policy leads to the following closed form solution of the individual rates of growth:*

$$R_i = \begin{cases} f^+, \ i = J, \\ -g^+, \ i \neq J, \end{cases} \quad a.s.$$

*where J is a random variable with values in $[N]$ whose exact value depends on $U(0)$, $f(\cdot)$, and $g(\cdot)$.*

*Proof of Theorem 8.* For $\forall i, j \in [N]$ s.t. $U_i(t) \geq U_j(t)$, we have $a_i(t) \geq a_j(t)$, then under the assumption that $f_i(x) \equiv f(x), g_i(x) \equiv g(x)$, we further obtain

$$\begin{aligned} \mathbb{E}[Z_i(t+1)] &= a_i(t) \cdot f(U_i(t)) - (1 - a_i(t)) \cdot g(U_i(t)) \\ &\geq a_j(t) \cdot f(U_i(t)) - (1 - a_j(t)) \cdot g(U_i(t)) \\ &\geq a_j(t) \cdot f(U_j(t)) - (1 - a_j(t)) \cdot g(U_j(t)) = \mathbb{E}[Z_j(t+1)]. \end{aligned}$$

where the last inequality holds because of modeling conditions (Assumption 2.(a), (b)). Consider $i \in \mathcal{M}_t$ where $\mathcal{M}_t = \arg\max_j\{U_j(t)\}$ and $i \in \mathcal{M}_t, j \in [N]$ such that $U_i(t) - U_j(t) \geq 1$, we have

$$\begin{aligned} &\mathbb{E}[U_i(t+1) - U_j(t+1) \mid \mathcal{F}_t] - (U_i(t) - U_j(t)) \\ &= \mathbb{E}[Z_i(t+1) - Z_j(t+1) \mid \mathcal{F}_t] \\ &= a_i(t) \cdot f(U_i(t)) - (1 - a_i(t)) \cdot g(U_i(t)) - (a_j(t) \cdot f(U_j(t)) - (1 - a_j(t)) \cdot g(U_j(t))) \\ &\geq a_i(t) \cdot f(U_i(t)) - (1 - a_i(t)) \cdot g(U_j(t)) - (a_j(t) \cdot f(U_i(t)) - (1 - a_j(t)) \cdot g(U_j(t))) \\ &= (a_i(t) - a_j(t)) \cdot f(U_i(t)) + (a_i(t) - a_j(t)) \cdot g(U_j(t)) \\ &\geq \frac{e^{M(t)} - e^{M(t)-1}}{\sum_{j \in [N]} e^{U_j(t)}} \cdot f^- + \frac{e^{M(t)} - e^{M(t)-1}}{\sum_{j \in [N]} e^{U_j(t)}} \cdot g^- \\ &\geq \frac{1 - e^{-1}}{N}(f^- + g^+) > 0. \end{aligned}$$

Now treat $U_i(t) - U_j(t)$ as the welfare process and apply adapted Lundberg's inequality (Lemma 3), we claim that with positive probability that $U_i(t) - U_j(t) \geq 1$ for $\forall t \geq 0$ when $U_i(0) - U_j(0) \geq 1$ where $i \in \mathcal{M}_0$. Then combine with the regularity condition (Assumption 3(c)), we have that with positive probability (lowerbounded by a constant) that $U_i(t) - U_j(t) \geq 1$ for $\forall t > 0$ where $i \in \mathcal{M}_0$. Then we apply the same reasoning for $j \in [N] \backslash i$ and conclude that with probability 1, the proportional max-U policy will fixate on one single individual asymptotically. $\qquad \square$

**Theorem 9.** *Under regularity (Assumption 3) and modeling conditions (Assumption 2.(a),(b)), the survival condition (Assumption 1), the proportional* min-U *policy leads to the following closed form solution of the individual rates of growth:*

$$R_i = \bar{\zeta}((f_1^+, \ldots, f_N^+), (g_1^-, \ldots, g_N^-)), \quad i = 1, \ldots, N, \quad a.s.$$

*Proof of Theorem 9.* The result can be proved by induction, and the proof of Theorem 3 applies here with minor modifications. We assume for $N-1$ individuals the conclusion holds, and consider $\mathcal{M} := \arg\max_j \{U_j(0)\}$ and $\mathcal{M}^c := [N] \backslash \mathcal{M}$. For $\forall l \in \mathcal{M}$,

$$a_l(t) \leq \frac{e^{-D(t)}}{1 + (N-1)e^{-D(t)}} \Rightarrow \sum_{i \in \mathcal{M}^c} a_i(t) \geq \frac{1}{1 + (N-1)e^{-D(t)}},$$

where $D(t) = \max_{j \in [N]} U_j(t) - \min_{i \in [N]} U_i(t)$. Hence there exists constant $C$ such that when $D(t) \geq C$, the survival condition for $\mathcal{M}^c$ is satisfied and we have

$$\bar{U}_{\mathcal{M}^c}(t+1) - \bar{U}_{\mathcal{M}^c}(t) = \sum_{i \in \mathcal{M}^c} w_i^{\mathcal{M}^c} a_i(t) \cdot f_i(U_i(t)) - (1 - a_i(t)) \cdot g_i(U_i(t))$$

$$\geq \sum_{i \in \mathcal{M}^c} w_i^{\mathcal{M}^c} a_i(t) \cdot f_i^- - (1 - a_i(t)) \cdot g_i^+$$

$$= \left( \sum_{i \in \mathcal{M}^c} a_i(t) - \sum_{j \in \mathcal{M}^c} \frac{g_j^+}{f_j^- + g_j^+} \right) \cdot \left( \sum_{k \in \mathcal{M}^c} \frac{1}{f_k^- + g_k^+} \right)^{-1}$$

$$\geq \left( \frac{1}{1 + (N-1)e^{-C}} - \sum_{j \in \mathcal{M}^c} \frac{g_j^+}{f_j^- + g_j^+} \right) \cdot \left( \sum_{k \in \mathcal{M}^c} \frac{1}{f_k^- + g_k^+} \right)^{-1} > 0$$

where $\bar{U}_{\mathcal{M}^c}(t)$, $w_i^{\mathcal{M}^c}$ are defined as in equation 5 for set $\mathcal{M}^c$. Hence we apply the conclusion for $\mathcal{M}^c$ and claim that there exists constant $T_{\mathcal{M}^c}$ such that when $\sum_{i \in \mathcal{M}} a_i(t) \leq \frac{1}{1 + (N-1)e^{-C}}$ for $\forall t \geq 0$, we have

$$\mathbb{E}\left[ \min_{j \in \mathcal{M}^c} U_j(t) \right] \geq \min_{j \in \mathcal{M}^c} +1, \quad \forall t \geq T_{\mathcal{M}^c},$$

$$\mathbb{E}\left[ \max_{j \in \mathcal{M}^c} U_j(t) \right] \leq \max_{j \in \mathcal{M}^c} -1, \quad \forall t \geq T_{\mathcal{M}^c}.$$

As for $i \in \mathcal{M}$,

$$\mathbb{E}[Z_i(t+1) \mid a_i(t), \mathcal{F}_t] \leq a_i(t) f_i^+ - (1 - a_i(t)) g_i^-$$

$$\leq \frac{1}{N - 1 + e^{-D(t)}} f_i^+ - \left( \frac{N - 2 + e^{-D(t)}}{N - 1 + e^{-D(t)}} \right) g_i^-,$$

and when $D(t) \geq C'$ for constant $C' > 0$, we have

$$\frac{1}{N - 1 + e^{-D(t)}} f_i^+ - \left( \frac{N - 2 + e^{-D(t)}}{N - 1 + e^{-D(t)}} \right) g_i^- < -\frac{1}{2} \min_{i \in [N]} g_i^-. \tag{31}$$

Hence for the whole population $[N]$, if $\sum_{i \in \mathcal{M}_t} a_i(t) \leq \min \left\{ \frac{1}{1 + (N-1)e^{-C}}, \frac{1}{2} \min_{i \in [N]} \frac{g_i^-}{f_i^+ + g_i^-} \right\}$, there exists constant $T$ such that

$$\mathbb{E}\left[ \min_{j \in \mathcal{M}^c} U_j(t) \right] \geq \min_{j \in \mathcal{M}^c} U_j(0) + 1, \quad \forall t \geq T,$$

$$\mathbb{E}\left[ \max_{j \in \mathcal{M}} U_j(t) \right] \leq \max_{j \in \mathcal{M}} U_j(0) - 1, \quad \forall t \geq T.$$

The rest of the proof goes through with minor modifications given the above facts. $\qquad \square$

## G    MODELING CHOICE ON MONOTONICITY

Use the assumption on the Matthew effect, we model $(f_i(\cdot))_i$ as increasing and $(g_i(\cdot))_i$ as decreasing. For other combinations of monotonicities (see Table 1), one could develop theoretical foundation using similar tools. We provide finite-horizon simulations for different combinations of monotonicities (as in Table 1) in Figure 5.

| $g_i(\cdot)$ \\ $f_i(\cdot)$ | Decreasing | Increasing | Constant |
|---|---|---|---|
| Decreasing | Mixed (Rawlsian) | **Rawlsian** | Mixed (Rawlsian) |
| Increasing | Mixed (utilitarian) | Utilitarian | Utilitarian |
| Constant | Tie | Tie | Tie |

Table 1: Here "increasing" denotes non-constant increasing functions and "decreasing" denotes non-constant decreasing functions. Each cell represent the comparison between the utilitarian policies and the Rawlsian policies under the monotonicicy combination of $f_i(\cdot)$, $g_i(\cdot)$ being "column-row" using the measure of long-term social welfare. We use "Mixed (utilitarian)" to represent regions where only utilitarian policies achieve the better long-term social welfare, "Mixed (Rawlsian)" to represent regions where all of Rawlsian policies and only part of utilitarian policies achieve the better long-term social welfare, "Utilitarian" to represent the region (discussed in this paper) where all of utilitarian policies show superiority over all of Rawlsian policies, "**Rawlsian**" to represent the region (discussed in this paper) where all of Rawlsian policies show better welfare than all utilitarian policies, "Tie" to represent regions where utilitarian policies and Rawlsian policies perform equally well. The max-g policy breaks the tie by choosing the individual with lowest welfare and the other policies break the tie by choosing the individual with the smallest index. Our discussion in Section 3 is constrained to cell on the first row, second column because of our assumption about "rich-get-richer" and "poor-get-poor". All the simulations are conducted under the survival condition and the uniform boundedness assumption.

From Figure 5, we observe the trend shifts as the monotonicity of decay functions $g_i(\cdot)$ change. As the monotonicity of decay functions describe where the instability of the society is located: decreasing $g_i(\cdot)$ stands for individuals are more fragile as their welfare level are lower while increasing $g_i(\cdot)$ implies more fragility of individuals with higher welfare level. Hence Rawlsian policies, which aim to leave no one behind, would perform better under cases where $g_i(\cdot)$ are decreasing. As when $g_i(\cdot)$ are constant (no inequality in decay functions), both types of policies will perform well show no difference in long-term welfare.

## H    POLICY COMPARISON FOR HETEROGENEOUS POPULATIONS

In this section, we discuss the comparisons among policies when the uniform boundedness assumption is violated and provide evidence that a Rawlsian policy still prevails over a utilitarian policy when the heterogeneity of the population is bounded below some threshold.

When the uniform boundedness condition may not hold, a direct comparison between Rawlsian and utilitarian policies becomes more intricate. We explore this case by simulating the long-term social welfare values when the limits of the intervention return and decay functions $f_i, g_i$ are different. We use the same dataset and pre-processing procedure as Section 5.1 except we generate heterogeneous bounds $\{f_i^-, f_i^+, g_i^-, g_i^+\}$ and conduct comparison using finite-horizon simulation.

We draw the bounds $g_i^-, g_i^+, f_i^-, f_i^+$ from normal distributions in the following way:

- The variance of the normal distribution is modeled by a parameter $\sigma^2$ that controls the heterogeneity of the bounds: larger $\sigma$ means more heterogeneous bounds.

- The mean of the normal distribution is chosen differently for $f_i$ and $g_i$: first of all, we choose means for the $f_i$ and $g_i$ functions that makes survival condition possible. Second, we introduce a parameter $b$ that models the strength of the decay functions: larger $b$ means that $g_i^-$ and $g_i^+$ are closer, and therefore, the decay effect is bounded. A smaller $b$ means

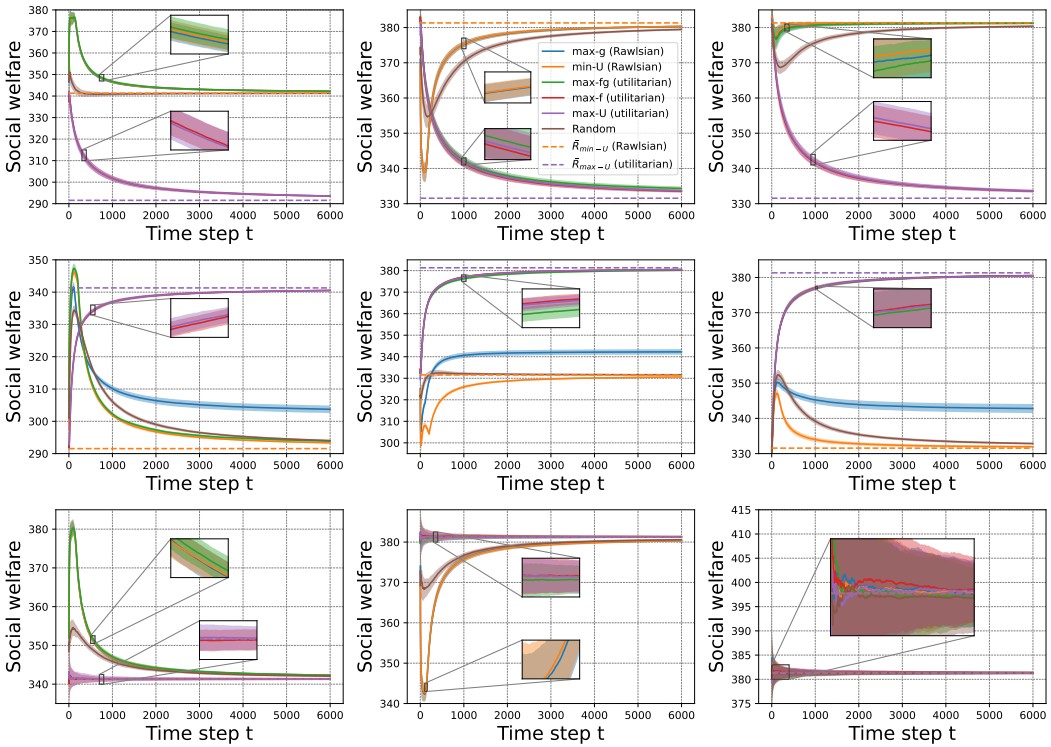

Figure 5: Social welfare as the finite-time growth rate averaged over all individuals, for all policies (solid lines), as well as theoretical growth rate for min-U and max-U policies (dashed lines) under different combination of monotonicities of $\{f_i(\cdot)\}$, $\{g_i(\cdot)\}$. The monotonicities of $\{f_i(\cdot)\}$, $\{g_i(\cdot)\}$ corresponds to Table 1, e.g., top left plot is generated for decreasing $\{f_i(\cdot)\}$ and $\{g_i(\cdot)\}$.

that the (relative) decay effect can get quite large for individuals with lower welfare, i.e., we expect a stronger "poor-get-poorer" effect.

Based on this set-up, we simulate the following model: w.l.o.g., we set $M = 1$, for each individual $i \in [N]$, $g_i^- \sim \mathcal{N}\left(bg^+, b^2\sigma^2\right)$, $g_i^+ \sim \mathcal{N}\left(g^+, \sigma^2\right)$, $f_i^- \sim \mathcal{N}\left(f^-, (\frac{f^-}{g^+})^2\sigma^2\right)$, $f_i^+ \sim \mathcal{N}\left(f^+, (\frac{f^+}{g^+})^2\sigma^2\right)$ where $g^+, f^-, f^+$ are constant parameters, $b \in (0, 1]$, $\sigma^2 > 0$. For each pair of $(b, \sigma^2)$ parameters, we generate 50 sets of heterogeneous bounds $(\{g_i^-, g_i^+, f_i^-, f_i^+\}_{i=1:N})$. Under each set of heterogeneous bounds $(\{g_i^-, g_i^+, f_i^-, f_i^+\}_{i=1:N})$, we average the social welfare obtained over all individuals, averaging over 50 iterations of the generation process of the intervention and decay function bounds. We present the finite-time social welfare of individuals under different policies using one set of randomly generated bounds $\{f_i^-, f_i^+, g_i^-, g_i^+\}$ in Figure 6. Figure 7 illustrates a heatmap of the percentage of times when the min-U Rawlsian policy has better long-term welfare than the max-U utilitarian policy, for each value of $b$ and $\sigma$ (1 meaning that the min-U Rawlsian policy is better in all iterations). From Figure 7, we observe the min-U Rawlsian policy maintains the tendency to perform worse the max-U policy in a short-term while surpasses the max-U utilitarian policy in the long-term for a range of $\sigma$ values (in a sense, for bounded heterogeneity). As $b$ decreases (and therefore the decay functions $g_i$ have a stronger effect), a Rawlsian policy starts performing better by preventing stronger loss caused by the decay of low-welfare individuals.

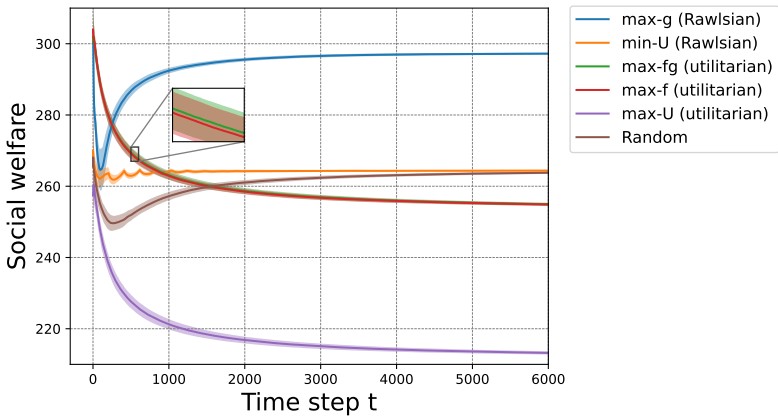

Figure 6: Social welfare as the finite-time growth rate average over all individuals, for all policies (solid lines) under one set of heterogeneous bounds.

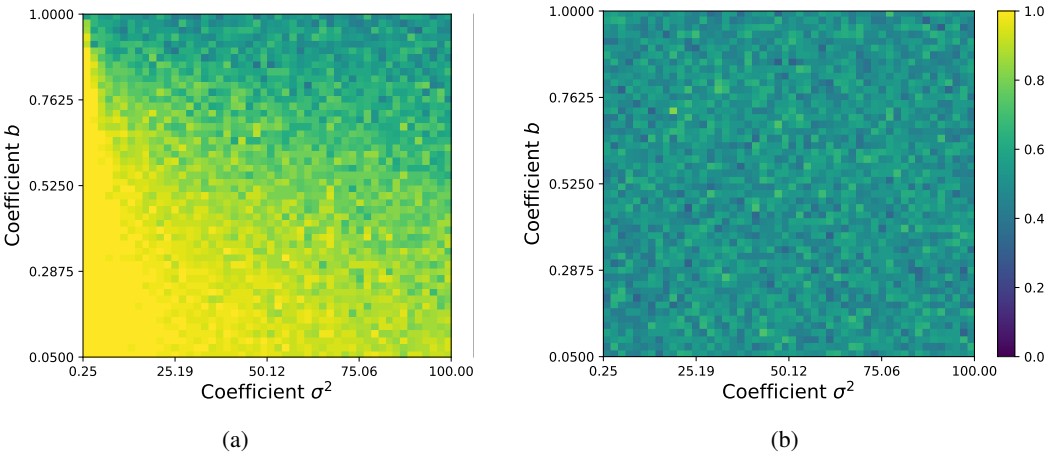

Figure 7: Percentage of iterations where min-U obtains better social welfare than max-U as the bounds $\{f_i^-, f_i^+, g_i^-, g_i^+\}_{i=1:N}$ vary according to parameters $b$ and $\sigma^2$, at time step $t = 6000$ (a) and $t = 10$ (b).

