# OpenReview forum: "Policy Design in Long-run Welfare Dynamics"
_ICLR.cc/2025/Conference — ICLR 2025 Poster_

### Official Review · Reviewer_PGAD · 2024-11-01

**Soundness:** 3
**Presentation:** 3
**Contribution:** 3
**Rating:** 6
**Confidence:** 2

**Summary:**

In this paper, the authors study how policies can affect long-term welfare in a population. Specifically, they prove that under the survival condition, the Rawlsian policy has greater long-run utility than the utilitarian policy, and under the ruin condition, the utilitarian policy has greater long-run utility than the Rawlsian policy. They demonstrate their findings through numerical simulations.

**Strengths:**

* The paper is very well-written and easy to follow.

* The paper studies the problem of how policies can affect the long-run welfare of the population, which is important and interesting.

* The paper presents theoretical results that have clear implications. The proof is technically sound.

However, since I have no background in economics, I find it challenging to evaluate the significance of the paper's contributions or the reasonableness of its welfare model. So I choose to assign a positive rating to the paper with a low confidence score.

**Weaknesses:**

See the "Questions" part.

**Questions:**

* In the welfare model it is assumed that everyone's welfare will decay over time without interventions. But why is this the case? In the paper [Roy Radner and Michael Rothschild, 1975], such a model is used to describe how a manager would assign his efforts to several tasks. In this circumstance, it is reasonable to have a "decay with time" assumption since if a task is ignored then things may get worse over time. However, I cannot see why such an assumption can be generalized to the model of welfare. Also, the "rich-get-richer" assumption seems at odds with the bounded $f$, $g$ assumption. I think the authors should add more explanations about the modeling choice to make the paper more accessible to readers outside the field. I also recommend the authors use real-world data or existing literature to rationalize such modeling choices. (In the current simulations only the initial state of the model is based on true data, and all other components are hand-crafted.)

* In the paper the author only considers the policy that focuses on one individual at a time. Can the authors comment on what the main technical difficulty is in considering a more general policy that forms a probability distribution on the population?

* In the paper, the authors do not touch on the problem of optimality of policies. It seems the current welfare model can be conceived as an average-reward continuous-state MDP. Since the functions $f_i$ and $g_i$ are assumed to be known, finding the optimal policy is equivalent to solving a planning problem in such an MDP, which is well-studied in the literature. Can the authors give a concise discussion about that?

---

> ### Author Response · Authors · 2024-11-20
>
> > In the welfare model it is assumed that everyone's welfare will decay over time without interventions. But why is this the case? In the paper [Roy Radner and Michael Rothschild, 1975], such a model is used to describe how a manager would assign his efforts to several tasks. In this circumstance, it is reasonable to have a "decay with time" assumption since if a task is ignored then things may get worse over time. However, I cannot see why such an assumption can be generalized to the model of welfare.
>
> Welfare studies have shown evidence that the "poor-get-poorer" through income shocks, health concerns, consumption fluctuations etc [1],[2],[3]. Our model focuses on cases where interventions are designed for susceptible populations to income shocks. These groups are often the target of resource allocation in policy design [4].
>
>
> Empirical studies show that income shocks have a vastly larger impact on consumption and savings behavior for lower-wealth individuals (see [5] and references therein). Furthermore, such studies also show that lower wealth individuals experience more frequent income shocks, and that those income shocks aggregate in a non-linear way, with an increased effect on consumption and savings for lower-asset groups. Instead of modeling income and consumption separately, we opted for a simpler model in which income shocks are modeled through a function $g_i$ that increases with lower welfare values. Similarly, extensive literature shows an increased value for in-kind transfers, cash transfers, and redistributive policies (e.g. taxation policies) for wealthier individuals as compared to poorer individuals (see [6],[7], and references therein). As such, we model interventions through the $f_i$ with an increased positive effect on the welfare values of wealthier individuals by considering monotonically increasing functions in the welfare value. We would be glad to include a more extensive discussion on the modeling choices in the final version of our paper.
>
>
> > Also, the "rich-get-richer" assumption seems at odds with the bounded $f,g$,  assumption. I think the authors should add more explanations about the modeling choice to make the paper more accessible to readers outside the field. I also recommend the authors use real-world data or existing literature to rationalize such modeling choices. (In the current simulations only the initial state of the model is based on true data, and all other components are hand-crafted.)
>
> We think there might be a misunderstanding here: the rich-get-richer effect is captured by modeling $f_i$ as an increasing function in the welfare, meaning that the richer an individual is, the higher the rate of return on welfare is. $f_i$ models the potential growth rate of an individual between two consecutive timesteps: an individual's welfare can indeed be unbounded as time progresses, but their growth rate between two consecutive timesteps is bounded.
>
>
>
> Our work is a first theoretical formulation that assesses the long-term comparison of policies. Assessing such a comparison with real-data requires extensive field experimentation, which is beyond the scope of our paper. Data on long-term effects of policy designs is rare, when it exists. As such, our simulations serve the purpose of illustrating our theoretical results by starting with a real-world income distribution.
>
> > In the paper the author only considers the policy that focuses on one individual at a time. Can the authors comment on what the main technical difficulty is in considering a more general policy that forms a probability distribution on the population?
>
> In Appendix E, we extend our model to a multi-budget framework that allows for focusing on multiple individuals at each time step, while our results are maintained qualitatively. Moreover, when resources are allocated based on a probability distribution over the population, it is equivalent to proportional budget allocation. Within this framework, the design of Rawlsian and utilitarian policies requires careful evaluation. We provide preliminary results on proportional resource allocation using example proportional policies in supplementary material.
> For the proportional min-U policy, our analysis techniques remain valid with non-essential modifications. But for proportional max-U policy, the analysis is more complex. While our earlier analysis of max-U policy demonstrated convergence to a single individual almost surely, this convergence analysis no longer holds under the proportional framework.

---

> ### Author Response · Authors · 2024-11-20
>
> > In the paper, the authors do not touch on the problem of optimality of policies. It seems the current welfare model can be conceived as an average-reward continuous-state MDP. Since the functions $f_i$ and $g_i$ are assumed to be known, finding the optimal policy is equivalent to solving a planning problem in such an MDP, which is well-studied in the literature. Can the authors give a concise discussion about that?
>
>
> We appreciate this point: we had included a short discussion in the related work section, containing relevant papers that tackle the problem of optimal policy through MDP-type methods [8],[9],[10]. We would be glad to extend this discussion with additional comparisons to relevant works. Concisely, the problem of optimal policy selection can indeed be solved through a continuous-state MDP under the average-reward criteria, with early works considering bounded reward rates [11] (like in our case) and subsequent extensions that do not require boundedness [12]. These works find theoretical guarantees for the *existence* of optimal policies, convergence rates, as well as optimality gaps. Often, such works do not find tractable, closed-form solutions for the optimal policy, but rather build heuristics with theoretical guarantees that can closely approximate an optimal policy. It makes the comparison between different policies difficult in theory.
>
>
> In contrast, our work finds **sufficient** theoretical conditions for the comparison of different policies that incorporate normative values (e.g. a Ralwsian approach vs a utilitarian approach). In other words, our work differs in giving general conditions on the state of the world through the survival and ruin conditions that allow a social planner to directly compare policies. Whereas the existence of an optimal policy may be known using an MDP-type method, a different instantiation of the model (with different parameters) may result in a different optimal policy for targeting, thus making a general policy comparison intractable. We would be glad to include an extended discussion on related work on optimal policy selection using continuous-state MDPs in the final version of our paper.
>
>
> [1] Abebe, R., Kleinberg, J., & Weinberg, S. M. (2020, April). Subsidy allocations in the presence of income shocks. In Proceedings of the AAAI Conference on Artificial Intelligence (Vol. 34, No. 05, pp. 7032-7039).
>
> [2] Durham, Y., Hirshleifer, J., & Smith, V. L. (1998). Do the rich get richer and the poor poorer? Experimental tests of a model of power. The American Economic Review, 88(4), 970-983.
>
> [3] Jiang, L., & Probst, T. M. (2017). The rich get richer and the poor get poorer: Country-and state-level income inequality moderates the job insecurity-burnout relationship. Journal of Applied Psychology, 102(4), 672.
>
> [4] Banerjee, A., Duflo, E., Goldberg, N., Karlan, D., Osei, R., Parienté, W., ... & Udry, C. (2015). A multifaceted program causes lasting progress for the very poor: Evidence from six countries. Science, 348(6236), 1260799.
>
> [5] Arellano, M., Blundell, R., Bonhomme, S., & Light, J. (2024). Heterogeneity of consumption responses to income shocks in the presence of nonlinear persistence. Journal of Econometrics, 240(2), 105449.
>
> [6] Warwick, R., Harris, T., Phillips, D., Goldman, M., Jellema, J., Inchauste, G., & Goraus-Tańska, K. (2022). The redistributive power of cash transfers vs VAT exemptions: A multi-country study. World Development, 151, 105742.
>
> [7] Bourguignon, F. (2018). Redistribution of Income and Reducing Economic Inequality-IMF F&D Magazine. IMF.
>
> [8] Zimmer, M., Glanois, C., Siddique, U., & Weng, P. (2021, July). Learning fair policies in decentralized cooperative multi-agent reinforcement learning. In International Conference on Machine Learning (pp. 12967-12978). PMLR.
>
> [9] Zhou, A. (2024). Optimal and fair encouragement policy evaluation and learning. Advances in Neural Information Processing Systems, 36.
>
> [10] Zheng, S., Trott, A., Srinivasa, S., Parkes, D. C., & Socher, R. (2022). The AI Economist: Taxation policy design via two-level deep multiagent reinforcement learning. Science advances, 8(18), eabk2607.
>
> [11] Doshi, B. T. (1976). Continuous time control of Markov processes on an arbitrary state space: average return criterion. Stochastic Processes and their Applications, 4(1), 55-77.
>
> [12] Guo, X., & Rieder, U. (2006). Average optimality for continuous-time Markov decision processes in Polish spaces.

---

> > ### Comment · Reviewer_PGAD · 2024-11-22
> >
> > Thanks for the response. I choose to retain my positive rating of this paper.

---

### Official Review · Reviewer_Zhdw · 2024-11-01

**Soundness:** 4
**Presentation:** 4
**Contribution:** 4
**Rating:** 8
**Confidence:** 5

**Summary:**

The paper studies sequential policy design with the goal of maximising long-term welfare. In particular the authors compare two policy classes—Rawlsian and utilitarian—and identify conditions under which each is optimal. They model the population as N agents, where each agent’s welfare naturally decays without intervention but improves with intervention. Sections 3 and 4 compare the policy classes in terms of long-term population welfare and individual welfare, respectively. Under regularity and survival conditions, the Rawlsian policy outperforms the utilitarian; however, under the ruin condition, this implication is reversed. Section 5 contains experiments with initial welfares drawn from SIPP data and the social welfare evolution supports the theory.

**Strengths:**

- The paper is well-structured and theoretically rigorous. Theorems 1 and 2 in Section 3 are particularly insightful, establishing necessary and sufficient conditions under which the Rawlsian and utilitarian policies are optimal.
- The individual welfare evolution and policy optimality for Rawlsian, utilitarian, and random policies in Section 4 is also novel, requiring new sub-martingale techniques in the proofs.
- Section 5 includes experimental results that support the theoretical finding and we see that Rawlsian policies though initially leading to a drop, are optimal for long-term social welfare.

**Weaknesses:**

- The modelling conditions, such as "rich-get-richer" and "poor-get-poorer," are well-explained, but the survival condition (line 255) is difficult to interpret. Is this condition standard from Radner and Rothschild, or is it specific to your framework?
- Clarifying how common the survival condition is would help, as knowing whether the system is in a survival or ruin state is essential to determining whether a Rawlsian or utilitarian policy is optimal.

**Questions:**

- Please address the questions above.
- A minor question: your theory seems to extend to interventions that allocate resources to more than one agent at each time step $\sum a_i(t) = M$. Is there a reason not to include this in the main body, with \(M=1\) as a special case?

---

> ### Author Response · Authors · 2024-11-20
>
> > The modelling conditions, such as "rich-get-richer" and "poor-get-poorer," are well-explained, but the survival condition (line 255) is difficult to interpret. Is this condition standard from Radner and Rothschild, or is it specific to your framework?
> Clarifying how common the survival condition is would help, as knowing whether the system is in a survival or ruin state is essential to determining whether a Rawlsian or utilitarian policy is optimal.
> Questions:
> Please address the questions above.
>
> We thank the reviewer for the positive feedback and thoughtful questions!
>
> The survival condition is inherited from the work of Radner and Rothschild and describes a state of the world in which proper policy would be able to ``save'' all individuals (aka no individual would reach negative welfare). An extreme example would be a model in which everyone starts with positive welfare, the decay function is constant and equal to 0, and the return function is non-negative. In this case, under any policy, no individual would achieve a negative welfare and then the survival condition is satisfied. Of course, this is just an idealized example to provide some intuition. In reality, the model parameters may be more complex and assessing whether a policy is effective is complicated: as such, the survival and ruin conditions provide a closed-form expression for this assessment and have a meaningful mathematical simplicity: they are a natural way of comparing policies. More specifically, the survival and ruin conditions allow us to compare Rawlsian and utilitarian policies w.r.t. the long-term average welfare. Whereas the survival condition can be characterized in closed-form as a function of the return and decay bounds, assessing whether we are currently in a state of survival would require some *a priori* knowledge or estimates of such bounds.
>
> Estimates of these bounds can be obtained by a social planner through pilot experiments, often done when designing social policies [1],[2], with experimentation rapidly increasing as a method for policy design and evaluation [3],[4],[5]. Thus, estimating the return and decay function bounds through small RCTs does not require a commitment to a particular policy, whose long-term effects are difficult to measure in practice. Hence, we think that the scope of our paper is to provide a mathematical theory in a simple yet rich model that can assess long-term policy outcomes. We believe that it would be in scope of more applied future work to combine effective estimation methods with policy assessment.
>
> > A minor question: your theory seems to extend to interventions that allocate resources to more than one agent at each time step . Is there a reason not to include this in the main body, with (M=1) as a special case?
>
> We thought of keeping the main analysis as simple and clear as possible, given all different parts of the model. But, we would be glad to include this extension in the main paper if the reviewer thinks it would be valuable.
>
> [1] Cabinet Office. (2003). Trying it Out: The Role of ‘Pilots’ in Policy Making. Report of a review of government pilots.
>
> [2] Huitema, D., Jordan, A., Munaretto, S., & Hildén, M. (2018). Policy experimentation: core concepts, political dynamics, governance and impacts. Policy Sciences, 51, 143-159.
>
> [3] Banerjee, A. V., Duflo, E., & Kremer, M. (2016). The influence of randomized controlled trials on development economics research and on development policy. The state of Economics, the state of the world, 482-488.
>
> [4] Banerjee, A., Banerji, R., Berry, J., Duflo, E., Kannan, H., Mukerji, S., ... & Walton, M. (2017). From proof of concept to scalable policies: Challenges and solutions, with an application. Journal of Economic Perspectives, 31(4), 73-102.
>
> [5] Webber, S., & Prouse, C. (2018). The new gold standard: The rise of randomized control trials and experimental development. Economic Geography, 94(2), 166-187.

---

> > ### Comment · Reviewer_Zhdw · 2024-11-25
> >
> > Thanks for your response. I’d suggest adding a discussion on survival and ruin in Section 6 and outlining how you plan to address this in future work. I also think it would be great to include the results for the multi-budget case (M) in the main text, as long as it fully generalizes the M=1 case.

---

> > > ### Author Response · Authors · 2024-11-26
> > >
> > > We greatly appreciate your thoughtful suggestions and valuable input! We will incorporate our discussions into the final version of our paper.

---

### Official Review · Reviewer_C3iz · 2024-11-04

**Soundness:** 3
**Presentation:** 4
**Contribution:** 3
**Rating:** 6
**Confidence:** 2

**Summary:**

The paper compares the long-term social welfare resulting from two popular policy frameworks, a Rawlsian policy that focuses on the minimizing the smallest welfare loss and a utalitarian policy that focuses on maximizing immediate welfare gain. The comparison is conducted under a model on welfare decay and intervention return, assuming a survival condition and the Matthew effect. The conclusions are that 1) when the survival condition is satisfied, the Rawlsian policy achieves better long-term social welfare than the utalitarian policy; and 2) when the ruin condition is satisfied, the utalitarian policy becomes better.

**Strengths:**

The paper puts the welfare policy evaluation in a sequential decision making framework. All models and assumptions are written clearly. The theoretical results are backed with numerical experiments and mathematical proof.

**Weaknesses:**

The theoretical results rely on several strong assumptions, and need more clarification.

**Questions:**

## Major comments:

1. I am curious about whether the two essential assumptions for the theoretical results in the paper are realistic. These assumption include 1) the return and decay functions are monotone, 2) the bounds on the return and decay functions are uniform across all individuals. For assumption 1), the paper mentions that the social planner does not need knowledge of $f_i$ or $g_i$. In this case, how does the social planner know whether $f_i$ and $g_i$ are monotone in practice? For assumption 2), the "rich-get-richer" modeling condition also needs to be satisfied. However in reality, the rich gets richer at a much faster rate than other individuals. Is assuming the same upper bound for all indivuals realistic?

2. What is the motivation and advantage for considering max-f and max-g policies, as opposed to the max-fg policy? The paper mentions that the max-d policy only requires partial information about the interventions, which is less costly to measure compared to measuring both return and decay. However, the max-g policy still requires information about the decays, which then offsets the previously mentioned advantage.

3. As for the survival condition, how should a social planner assess whether this condition is satisfied in practice?


## Minor comments:

1. Equation (1): The definition of $\mathcal{F}_t$ appears much later.

---

> ### Author Response · Authors · 2024-11-20
>
> > Major comments:
> I am curious about whether the two essential assumptions for the theoretical results in the paper are realistic. These assumption include 1) the return and decay functions are monotone, 2) the bounds on the return and decay functions are uniform across all individuals. For assumption 1), the paper mentions that the social planner does not need knowledge of $f_i$ or $g_i$. In this case, how does the social planner know whether $f_i$ and $g_i$ are monotone in practice?
>
> We thank the reviewer for their positive feedback and detailed questions. In particular, we agree that a more detailed discussion and motivation of the assumptions would strengthen the paper. For instance, regarding the monotonicity of $f_i, g_i$, these properties serve the purpose of capturing known general social trends such as the rich-get-richer and poor-get-poorer effects often observed in income fluctuations studies. For example, empirical studies show that income shocks have a vastly larger impact on consumption and savings behavior for lower-wealth individuals (see [1] and references therein). Furthermore, such studies also show that lower wealth individuals experience more frequent income shocks, and that those income shocks aggregate in a non-linear way, with an increased effect on consumption and savings for lower-asset groups. Instead of modeling income and consumption separately, we opted for a simpler model in which income shocks are modeled through a function $g_i$ that increases with lower welfare values. Similarly, extensive literature shows an increased value for in-kind transfers, cash transfers, and redistributive policies (e.g. taxation policies) for wealthier individuals as compared to poorer individuals (see [2], [3], and references therein). As such, we model interventions through the $f_i$ with an increased positive effect on the welfare values of wealthier individuals by considering monotonically increasing functions in the welfare value. We would be glad to include a more extensive discussion on the modeling choices in the final version of our paper.
>
> Furthermore, our modeling choices generalize a much simpler model from [4] that assumes such functions to be constant. We provide an extension in Appendix D for when $f_i$ is not monotone, but rather concave.
>
> The social planner does not need to know the **exact** form of $f_i,g_i$ for each individual. Rather, they can estimate general trends and effects of income shocks and interventions through small pilot experiments (often considered necessary precursors of policy deployment [5],[6]) or through acquiring domain knowledge (e.g., through poverty trackers [7] or longitudinal studies of intervention effects on income [7]).
>
>
> > For assumption 2), the "rich-get-richer" modeling condition also needs to be satisfied. However in reality, the rich gets richer at a much faster rate than other individuals. Is assuming the same upper bound for all indivuals realistic?
>
> We think there might be a misunderstanding here: the rich-get-richer effect is captured by modeling $f_i$ as an increasing function in the welfare, meaning that the richer an individual is, the higher the rate of return on welfare is. Decay function $f_i$ models the potential growth rate of an individual between two consecutive timesteps: an individual's welfare can indeed be unbounded as time progresses (with the rich getting richer faster than others), but the growth rate between two consecutive timesteps is bounded.
>
> We believe that the bounded homogeneity assumption is less restrictive than it seems at first: individuals can have different growth rate functions $f_i$, as long as the supremum of the growth rate is bounded above by some value. In that way, we believe our model can cover a rich class of functions that model the return and decay of individuals. We agree that an analysis for even more general conditions, such as a fully heterogenous bounds case, would be an excellent avenue for future work.

---

> ### Author Response · Authors · 2024-11-20
>
> > What is the motivation and advantage for considering max-f and max-g policies, as opposed to the max-fg policy? The paper mentions that the max-d policy only requires partial information about the interventions, which is less costly to measure compared to measuring both return and decay. However, the max-g policy still requires information about the decays, which then offsets the previously mentioned advantage.
>
> The max-f and max-g policies are introduced as partial information policies: a social planner may be able to estimate either f or g from prior studies without knowledge of the other. In that sense, they require separate measurements, and knowing one of them does not imply that we also know the other. For example, a social planner may gather information about the decay function by measuring income shocks through observational data collected from job centers, employments offices, national surveys etc. On the other hand, a social planner may also measure the effect of interventions such as cash or in-kind subsidies by measuring the welfare increase and estimating the return function $f$. Thus, we introduced the partial information policies max-f and max-g for completeness, for the interested reader.
>
>
> > As for the survival condition, how should a social planner assess whether this condition is satisfied in practice?
>
> Assessing whether the survival condition is satisfied in practice is a good question for the application of our work: a social planner may obtain preliminary estimates for the bounds on return and decay functions through pilot studies. Then, these estimates can be used as plug-in estimates in the survival condition. Pilot experiments are often used prior to deployment policies at a larger-scale in a population [5],[6], with experimentation rapidly increasing as a method for policy design and evaluation [8],[9],[10]. We believe that it would be in the scope of an applied study to combine effective estimation methods with policy assessment, which is an excellent avenue for future work. We would be glad to include this discussion and references in the final version of our paper.
>
>
> > Minor comments:
> Equation (1): The definition of $\mathcal{F}_t$ appears much later.
>
> We thank the reviewer for the observation. We revised the manuscript with an added definition after equation $(1)$ and will integrate it in the final version of our paper.
>
>
>
> [1] Arellano, M., Blundell, R., Bonhomme, S., & Light, J. (2024). Heterogeneity of consumption responses to income shocks in the presence of nonlinear persistence. Journal of Econometrics, 240(2), 105449.
>
> [2] Warwick, R., Harris, T., Phillips, D., Goldman, M., Jellema, J., Inchauste, G., & Goraus-Tańska, K. (2022). The redistributive power of cash transfers vs VAT exemptions: A multi-country study. World Development, 151, 105742.
>
> [3] Bourguignon, F. (2018). Redistribution of Income and Reducing Economic Inequality-IMF F&D Magazine. IMF.
>
> [4] Radner, R., & Rothschild, M. (1975). On the allocation of effort. Journal of economic theory, 10(3), 358-376.
>
> [5] Cabinet Office. (2003). Trying it Out: The Role of ‘Pilots’ in Policy Making. Report of a review of government pilots.
>
> [6] Huitema, D., Jordan, A., Munaretto, S., & Hildén, M. (2018). Policy experimentation: core concepts, political dynamics, governance and impacts. Policy Sciences, 51, 143-159.
>
> [7] Garfinkel, I. (2021). New York City Longitudinal Survey of Well-Being (Poverty Tracker), 2015-2018.
>
> [8] Banerjee, A. V., Duflo, E., & Kremer, M. (2016). The influence of randomized controlled trials on development economics research and on development policy. The state of Economics, the state of the world, 482-488.
>
> [9] Banerjee, A., Banerji, R., Berry, J., Duflo, E., Kannan, H., Mukerji, S., ... & Walton, M. (2017). From proof of concept to scalable policies: Challenges and solutions, with an application. Journal of Economic Perspectives, 31(4), 73-102.
>
> [10] Webber, S., & Prouse, C. (2018). The new gold standard: The rise of randomized control trials and experimental development. Economic Geography, 94(2), 166-187.

---

### Official Review · Reviewer_DG4s · 2024-11-05

**Soundness:** 3
**Presentation:** 4
**Contribution:** 3
**Rating:** 6
**Confidence:** 4

**Summary:**

This paper studies the behavior of Rawlsian and utilitarian policies in long-run welfare dynamics. They show that under a survival condition, Rawlsian policies are better than the idealized utilitarian approach almost surely in the long run. Under a ruin condition, a utilitarian policy will achieve better long-term social welfare. Simulation results are provided to validate the theories.

**Strengths:**

This paper is in general well-written and smooth to follow. The theoretical results are strong and the technical proofs seem to be rigorous and well-organized. The simulation results validates the theories clearly.

**Weaknesses:**

The authors basically leave the survival and ruin conditions unjustified, and the simulation settings are ad-hoc. I would appreciate it if the authors could provide discussion and common example settings where the conditions hold.

**Questions:**

- In line 195, the author motivates max-U policy with the setting where $f_i + g_i$ are increasing functions. However, this is not enough, right? Say $f_1 + g_1$ and $f_2 + g_2$ are such that $u_1 < u_2$ but  $f_1(u_1) + g_1(u_1) \geq f_2(u_2) + g_2(u_2)  $ (although $f_1 + g_1$ and $f_2 + g_2$ themselves are increasing functions).
- In lines 218 and 219, why the max-g policy is considered a variant of Rawlsian policy?  I mean, $i=\underset{j}{\arg \max }\{g_j(U_j(t))\} $ is still maximizing, rather than minimizing the welfare.

---

> ### Author Response · Authors · 2024-11-20
>
> ### Weaknesses
> > The authors basically leave the survival and ruin conditions unjustified, and the simulation settings are ad-hoc. I would appreciate it if the authors could provide discussion and common example settings where the conditions hold.
>
> * We thank the reviewer for their detailed feedback and insightful questions. In particular, we appreciate the need for a detailed discussion on the survival/ruin conditions and simulation settings. Our theoretical analysis reveals sufficient conditions for the long-term comparison of policies, namely the survival and ruin conditions. These conditions describe two potential states of the world: under survival, there exists a policy that prevents all individuals from reaching negative welfare at any time with positive probability. Under the ruin condition, no policy is able to prevent all individuals from reaching negative welfare (e.g., bankruptcy) over time with positive probability. Both the survival and ruin condition are mathematically natural, as our closed-form expressions show. They also present a meaningful way for policy comparison: on the one hand, under the ruin condition, a Rawlsian policy is not competitive to a utilitarian policy. On the other hand, a Rawlsian policy is better on average, in the long run, than a utilitarian policy under the survival condition. In other words, the survival condition is a strong complement of the ruin condition and it provides a meaningful testbed for assessing the efficacy of a Rawlsian policy.
> * Simulation settings: our simulations serve the purpose of illustrating the theoretical results, as the reviewer also mentioned. For analyzing policies under finite-time horizon, we directly simulate our model from equation (1) with Gaussian noise; we chose the bounds the return and decay functions as segment linear functions to preserve their monotonicity properties. We also experimented with other monotonic functions, such as sigmoid functions, obtaining qualitatively very similar results. Additionally, our simulations from Section 5.2 serve the purpose of illustrating the case when the homogeneity assumption does not hold: in doing so, we drew heterogeneous bounds from Gaussian distributions with different means and variances. We find a general qualitative trend: more heterogeneity in the bounds leads to a greater advantage for utilitarian policies. We note that the decay and return function bounds may also be estimated from a real-world experiment, which is beyond the scope of our work.
>
>
> ### Questions:
> > In line 195, the author motivates max-U policy with the setting where $f_i+g_i$ are increasing functions. However, this is not enough, right? Say $f_1+g_1$ and $f_2+g_2$ are such that $u_1<u)2$ but $f_1(u_1)+g_1(u_1)\geq f_2(u_2)+g_2(u_2)$ (although $f_1+g_1$ and $f_2+g_2$ themselves are increasing functions).
>
> We thank the reviewer for this observation. We agree, in cases where $f_i + g_i$ varies by individual, it does not necessarily reduce to the max-U policy. The max-U policy is motivated by cases where a decision-maker may not have access to f or g without epxerimental data. We updated our manuscript to reflect this point.
>
> > In lines 218 and 219, why the max-g policy is considered a variant of Rawlsian policy? I mean, $i=\text{argmax}_j g_j(U_j(t))$ is still maximizing, rather than minimizing the welfare.
>
> Our `poor-get-poorer' assumption models $g_j(U_j(t))$ as a decreasing function (so, individuals with higher welfare suffer a smaller decay, whereas individuals with low welfare suffer a larger decay). Thus, maximizing the decay value $g_j(U_j(t))$ is equivalent to minimizing the welfare value $U_j(t)$.

---

> > ### Comment · Reviewer_DG4s · 2024-11-29
> >
> > I appreciate the authors' replies to my questions and the corresponding updates. I tend to keep my score as it reflects my overall opinion.

---

### Author Response · Authors · 2024-11-20

We thank all reviewers for their insightful and positive reviews!  We are glad that reviewers find our problem important and interesting (@Reviewer PGAD). We are encouraged that our theoretical results are appreciated as novel (@Reviewer Zhdw), strong (@Reviwer DG4s) and rigorous (@Reviewer Zhdw, PGAD). We are pleased to find our theoretical results to have clear implications (@Reviewer PGAD) and simulations to clearly validate our theory (@Reviewer DG4s). We are glad that all reviewers liked the presentation of our results. We address reviewer comments below and we will include these discussions in the final version of our paper.

---

### Comment · Reviewer_C3iz · 2024-11-22

I would like to thank the authors for the detailed responses and clarification. The responses have convinced me of the theoretical contributions of the paper, and there I have raised my score.

---

> ### Author Response · Authors · 2024-11-26
>
> We thank the reviewer for reading the rebuttal and positive response! A quick follow up that the reviewer mentioned they raised their score, but we cannot see the change on Openreview.

---

> > ### Comment · Reviewer_C3iz · 2024-11-26
> >
> > Thank you for checking in. I confirmed that I have raised my score on "Contribution" from 2 to 3.

---

### Meta-Review · Area_Chair_pWTH · 2024-12-20

**Metareview:**

This paper investigates a stochastic dynamic model of long-term welfare within a population, where individual welfare improves with intervention and deteriorates without treatment. The authors compare two fundamental policies: the utilitarian policy, which maximizes immediate welfare improvement by treating the individual offering the highest immediate gain, and the Rawlsian policy, which focuses on the individual with the lowest welfare. Contrary to common critiques of Rawlsian approaches, the authors demonstrate that, under certain conditions, the Rawlsian policy yields greater long-term utility than the utilitarian policy, despite being less effective in the short term.

Overall, the reviewers appreciated the paper's theoretical contributions and its exploration of a socially impactful topic. While some concerns were raised regarding the paper's assumptions and their alignment with real-world scenarios (especially the survival and ruin conditions and the monotonicity of the return and decay functions), for the most part these were addressed convincingly during the rebuttal phase. The semi-synthetic simulations, though somewhat limited in scope, were deemed sufficient to support the paper's theoretical claims, and the reviewers agreed that the theoretical insights and implications of this work were significant enough to outweigh the limitations. Overall, the reviewers found the paper to be a solid contribution to the study of welfare optimization and reached a consensus to recommend acceptance.

**Additional Comments On Reviewer Discussion:**

As per ICLR policy, I am repeating here the relevant part of the metareview considering the reviewer discussion.

> While some concerns were raised regarding the paper's assumptions and their alignment with real-world scenarios (especially the survival and ruin conditions and the monotonicity of the return and decay functions), for the most part these were addressed convincingly during the rebuttal phase. The semi-synthetic simulations, though somewhat limited in scope, were deemed sufficient to support the paper's theoretical claims, and the reviewers agreed that the theoretical insights and implications of this work were significant enough to outweigh the limitations. Overall, the reviewers found the paper to be a solid contribution to the study of welfare optimization and reached a consensus to recommend acceptance.

---

### Decision · Program_Chairs · 2025-01-22

Accept (Poster)